# Granger Causal Interaction Skill Chains

**Caleb Chuck**                                                                *calebc@cs.utexas.edu*
*University of Texas at Austin*

**Kevin Black**
*University of California Berkeley*

**Aditya Arjun**
*University of Texas at Austin*

**Yuke Zhu**
*University of Texas at Austin*

**Scott Niekum**
*University of Massachusetts Amherst*

**Reviewed on OpenReview:** *https://openreview.net/forum?id=iA2KQyoun1*

## Abstract

Reinforcement Learning (RL) has demonstrated promising results in learning policies for complex tasks, but it often suffers from low sample efficiency and limited transferability. Hierarchical RL (HRL) methods aim to address the difficulty of learning long-horizon tasks by decomposing policies into skills, abstracting states, and reusing skills in new tasks. However, many HRL methods require some initial task success to discover useful skills, which paradoxically may be very unlikely without access to useful skills. On the other hand, reward-free HRL methods often need to learn far too many skills to achieve proper coverage in high-dimensional domains. In contrast, we introduce the Chain of Interaction Skills (COInS) algorithm, which focuses on *controllability* in factored domains to identify a small number of task-agnostic skills that still permit a high degree of control. COInS uses learned detectors to identify interactions between state factors and then trains a chain of skills to control each of these factors successively. We evaluate COInS on a robotic pushing task with obstacles—a challenging domain where other RL and HRL methods fall short. We also demonstrate the transferability of skills learned by COInS, using variants of Breakout, a common RL benchmark, and show 2-3x improvement in both sample efficiency and final performance compared to standard RL baselines.

## 1 Introduction

Reinforcement learning (RL) methods have shown promise in learning complex policies from experiences on various tasks, from weather balloon navigation (Bellemare et al., 2020) to Starcraft (Vinyals et al., 2019). Nevertheless, they often struggle with high data requirements and brittle generalization in their learned controllers (Nguyen & La, 2019). One promising avenue to improve sample efficiency and generalization is by incorporating hierarchical skills.

To address the limitations of vanilla RL, hierarchical RL (HRL) methods exploit temporal and state abstractions. Standard "flat" RL methods learn a monolithic policy, while HRL constructs a temporal hierarchy in which higher-level policies invoke lower-level policies (also called *skills* or *options*) and the lowest-level skills invoke primitive actions. This structure offers three major benefits: First, skills can represent temporally extended behaviors, decomposing long-horizon tasks into a sequence of shorter temporal segments. Second, an appropriate selection of skills fosters useful state abstractions, reducing the state-space complexity by

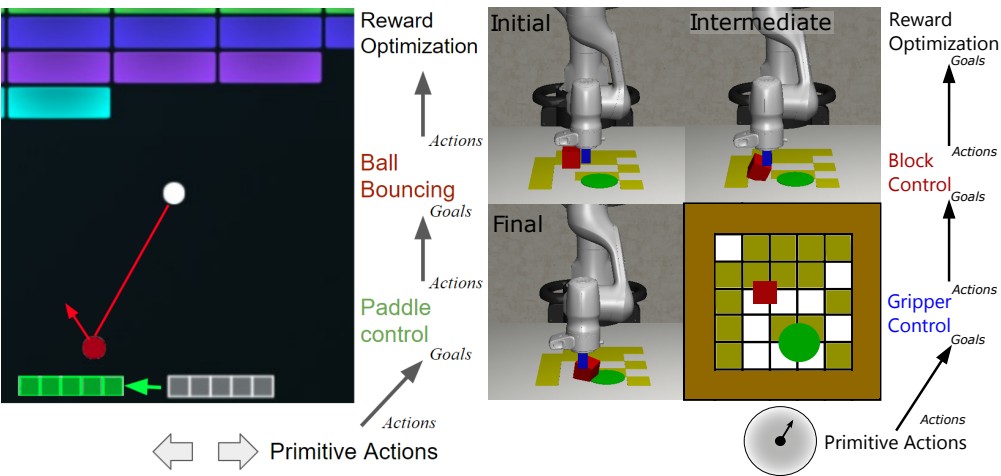

Figure 1: **Left:** The chain of COInS goal-based skills for Breakout, from primitive actions to final reward optimization. The goal space of the skills for one factor is the action space of the next factor controlling skill in the chain. COInS uses Granger-causal tests to detect interactions and construct edges between pairs of factors and their corresponding skills. **Right:** The Robot Pushing domain with negative reward regions. The objective is to push the block (red) from a random start position to a random goal (green) while avoiding the negative regions (shown in yellow, $-2$ reward), which are generated randomly in 15 grid spaces of a $5{\times}5$ grid over the workspace.

merging states through skill-based characteristics (*e.g.*, skill preconditions and/or effects). Third, the skills can be reused across different tasks to enable multi-task learning and fast adaptation (Kroemer et al., 2021).

Skills can be acquired through either reward-based methods or reward-free methods, each with challenges. Using reward for skill learning (Konidaris & Barto, 2009; Levy et al., 2019b) limits these HRL algorithms to update only when the agent achieves meaningful progress. For sparse-reward, long-horizon tasks, this often means that the learner flounders indefinitely until the first success is observed. On the other hand, many reward-free methods utilize state-covering statistics (Eysenbach et al., 2019). They struggle to learn in high-dimensional environments because learning skills that cover the entire state space is inefficient. Furthermore, even after learning those skills, the number of skills can end up excessively large for a high-level controller. While a handful of methods suggest ways to distinguish one state over another (Şimşek & Barto, 2008; Machado et al., 2017), this prioritization is an open research area.

While HRL methods have shown promising results, the existing limitations lead to low sample efficiency and hinder transferability. One reason for this is that neither novelty nor reward-based methods explicitly identify key states that mediate controllability. In many environments, these states, characterized by object interactions, bottleneck the high reward states: without reaching and performing precise control of these interaction states, the policy cannot perform well. For example, consider the game of Breakout (Figure 1 left) with the game state decomposed into a paddle, ball, and blocks. A strong policy in this domain can control each of these objects. However, learning this requires identification and credit assignment of rare and intermittent ball bounces. Similarly in the robot pushing scenario (Figure 1 right), the agent must learn complex manipulation of the block. Intuitively, manipulating interactions gives a policy a high degree of control, while failure to produce interactions often results in poor performance.

This intuition is drawn from the observation that human behavior is directed at causing *factor* interactions. In this work, we use "factor" to describe either a collection of features like the state of an object or non-state factors such as actions and rewards. Unlike reward-based skills, humans, even as young as infants, often exhibit exploratory behavior without obvious top-down rewards. Unlike novelty-based skills, this behavior is not state-covering but directed towards causing particular effects between factors (Rochat, 1989). The framework of directed behavior causing one factor to influence another decouples a small subset of interacting factors from the combinatorial state space of all factors.

The proposed Chain of Interaction Skills (COInS) algorithm identifies pairwise interactions in factored state spaces to learn a hierarchy of factor-controlling skills. COInS extends the Granger-causal test (Granger,

1969; Tank et al., 2021) with learned forward dynamics models and state-specific interactions. It uses Granger-causal relationships between factors to construct a chain of skills. Intuitively, Granger-causality determines if adding information about a new factor, which we refer to as the parent factor, is useful for predicting another factor, the "target" factor (see Figure 2). Then, goal-conditioned reinforcement learning learns interaction-controlling skills.

Using interactions, COInS automatically discovers a hierarchy of skills starting with primitive actions. These skills control progressively more difficult-to-control factors using already acquired skills, ending with reward optimization. In Breakout, COInS discovers the intuitive sequence of skills described in Figure 1. We demonstrate that the COInS algorithm not only learns the skills efficiently in the original version of the task but that the learned skills can transfer to different task variants—even those where the reward structure makes learning from scratch difficult. We show sample efficient learning and transfer in variations of the game Breakout and a Robot Pushing environment with high variation of obstacles, which are challenging domains for conventional RL agents.

Our work has three main contributions:

**1)** An unsupervised method of detecting interactions via an adapted Granger Causality criterion using learned forward dynamics models;
**2)** A skill-chain learning algorithm (COInS) driven by these discovered interactions; and
**3)** Empirical results demonstrating how COInS, a proof-of-concept instantiation of interaction-guided goal-based HRL, can sample efficiently learn transferable, high-performance policies in domains where skills controlling pairwise factors can achieve high performance.

## 2 Related Work

The literature on skill learning is often divided into two categories: reward-free (task-agnostic) skills and reward-based (task-specific) skills. Reward-free skills often utilize values derived from state visitation, such as information-theoretic skills (Eysenbach et al., 2019), to ensure that the skills cover as many states as possible while still being able to distinguish skills apart. Other reward-free skill learning uses tools such as the information bottleneck (Kim et al., 2021), transition Lagrangian (Machado et al., 2017) and dynamics (Sharma et al., 2020). While COInS also learns skills based on state information, it prioritizes controlling interaction-related states instead of arbitrary coverage. Interactions bear some resemblance to curiosity (Burda et al., 2018; Savinov et al., 2019) and surprise-based reward-free exploration methods (Berseth et al., 2021) in how it uses model discrepancy to set goals. COInS introduces Granger-causal interaction detection to learn reward-free skills that are not state-covering but capture the space of controllable factor relationships.

Another line of HRL methods backpropagate information from extrinsic rewards or goals (Barto & Mahadevan, 2003). Hierarchical Actor Critic (Levy et al., 2019b;a) uses goal-based rewards in locomotion tasks and bears similarities to COInS. Unlike COInS, it uses complete states as goals and does not scale to complex object manipulation where there is a combinatorial explosion of configurations. Other hindsight goal-based hierarchical methods propagate information from the true reward signal through distance metrics (Ren et al., 2019) or imitation learning (Gupta et al., 2019), neural architectures (Bacon et al., 2017), option switching (termination) cost (Harb et al., 2018), or variational inference (Haarnoja et al., 2018a). However these struggle with the same sparse, long-horizon reward issues. COInS shares similarities to causal dynamics learning (Wang et al., 2022; Seitzer et al., 2021), though these methods are non-hierarchical and focus on general causal relationships instead of interaction events. COInS builds a skill chain similar to HyPE (Chuck et al., 2020), which uses strong inductive biases with changepoint-based interactions. COInS employs fewer biases using a learned interaction detector and goal-based RL. In summary, COInS presents a novel approach to defining skills for hierarchical reinforcement learning leveraging interactions as a key mechanism.

## 3 Overview and Background

### 3.1 Overview

This work is motivated by an observation derived from the game Breakout: the agent in Breakout typically executes a lengthy series of 20 to 100 actions between the bounces of the ball. Identifying the precise sequence of actions that correlates with the reward from hitting a block often necessitates a large amount

of data, and the resulting policy may become overly tailored to the specific environment settings. However, by breaking down Breakout into a sequence of intermediate interactions—first between the actions and the paddle's position, then between the paddle and the ball, and finally between the ball and the block (similar to Figure 1a)—the task becomes significantly simpler. This work introduces a general strategy for decomposing control into intermediate, factor-based interactions that can be applied to a wide range of tasks, instantiated as Chain of Interaction Skills (COInS).

At each level of the hierarchy, COInS constructs a temporally abstracted MDP where the agent selects different interactions. For example, consider a version of Breakout where the action space consists of ball angles, and the state transitions occur at ball bounces. This abstraction trivializes the basic challenge of hitting blocks and transfers to more challenging versions of Breakout where the agent must strike specific blocks for positive reward.

This work aims to get this abstracted MDP by **(1)** identifying interactions, **(2)** training skills to produce those interactions, **(3)** building interaction skill chains from the bottom up, beginning with primitive actions and progressing to more complex factor relationships. In this section we formalize the chain of skills used in this abstracted MDP, starting from the Factored Markov Decision Process (FMDP). In Section 4.2 we introduce the Granger-causal method for detecting interactions, in Section 4.4 we formalize skill chain learning, and in Section 4.5 we describe how the skill chain is iteratively constructed.

### 3.2 Factored Markov Decision Processes

A Markov Decision Process (MDP), is described by a tuple $E := (\mathcal{S}, p, r, \mathcal{A}_{\text{prim}}, \gamma)$. $\mathcal{S}$ is the *state space*, $\mathcal{S}_\circ$ is the initial state space, $\mathbf{s} \in \mathcal{S}$ is a particular *state*, and $S$ denotes the *random variable* for state. We use script notation ($\mathcal{S}$) to represent sets, uppercase ($S$) for random variables, and boldface ($\mathbf{s}$) for vectors. The *primitive action space* $\mathcal{A}_{\text{prim}}$, which can be either discrete or continuous, comprises the available actions $\mathbf{a}_{\text{prim}} \in \mathcal{A}_{\text{prim}}$. In this work, we extend the factored MDP (FMDP) formulation (Boutilier et al., 1999; Sigaud & Buffet, 2013), where the state space is factorized into $n$ *factors*: $\mathcal{S} = \mathcal{S}_1 \times \ldots \times \mathcal{S}_n$. Each factor represents a distinct component of the overall state, such as the state of an individual object. Each state factor $\mathbf{s}_i \in \mathcal{S}_i$ is represented by a fixed length vector of real-valued *features*, denoted with $\mathbf{s}_i[k]$ for the feature indexed by $k$ of factor $\mathbf{s}_i$. For this work, we assume that the factorization is predefined, as this problem is being actively investigated in vision (Voigtlaender et al., 2019; Kirillov et al., 2023) and robotics (Lin et al., 2020) but is not the focus of this work. Our goal is to identify controllable state factors and the Granger-causal relationships between controllable factors.

The *transition function* is a probability distribution over $S'$, the next state, given $\mathbf{s}$ the current state, and $\mathbf{a}_{\text{prim}}$ the current primitive action. It is defined as $p : \mathcal{S} \times \mathcal{A}_{\text{prim}} \times \mathcal{S} \to P(\cdot | S = \mathbf{s}, A_{\text{prim}} = \mathbf{a}_{\text{prim}})$. In the FMDP formulation, the transition function is represented with a Bayesian network connecting state factors $\mathbf{s}_i$ and $\mathbf{a}_{\text{prim}}$ at the current time step with the next state factors $\mathbf{s}'_i$. Previous works utilize the sparsity of this network for efficient RL. In this work we extend this to state-conditional sparsity: in certain states, the connectivity is especially sparse, ex. when a ball is moving through free space in Breakout.

The *reward function* is a mapping from states, primitive actions, and next states to real-valued rewards: $r : \mathcal{S} \times \mathcal{A}_{\text{prim}} \times \mathcal{S} \to \mathbb{R}$. A *policy* is a function mapping states to the probability distribution over actions, such that $\pi : \mathcal{S} \to P(\cdot | S = \mathbf{s})$. A trajectory is a length $T$ sequence of state action pairs: $\tau := (\mathbf{s}^{(0)}, \mathbf{a}_{\text{prim}}^{(0)}, \mathbf{s}^{(1)}, \mathbf{a}_{\text{prim}}^{(1)}, \ldots \mathbf{s}^{(T-1)}, \mathbf{a}_{\text{prim}}^{(T-1)})$, The probability of a trajectory under a particular policy is $P_\pi(\tau) = P(S_\circ = \mathbf{s}^{(0)}) \prod_{t=0}^{T-1} \pi(\mathbf{a}_{\text{prim}}^{(t)} | \mathbf{s}^{(t)}) p(\mathbf{s}^{(t+1)} | \mathbf{s}^{(t)}, \mathbf{a}_{\text{prim}}^{(t)})$, where $S_\circ$ is the initial state distribution. We represent the trajectory distribution given a particular policy as $\rho(\pi)$. The objective of RL is to learn a policy that maximizes the expected *return*, the $\gamma$-discounted sum of rewards, for $\gamma \in [0, 1]$, defined as:

$$\text{ret}[\pi] = E_{\tau \sim \rho(\pi)} \left[ \sum_{t=0}^{T-1} \gamma^t r(\mathbf{s}^{(t)}, \mathbf{a}^{(t)}) \right]$$

### 3.3 Skills

This work builds on the skills/options framework described by the semi-MDP (sMDP) (Sutton et al., 1999) formulation. A *skill* or option in an sMDP is defined by the tuple $\omega := (\mathcal{I}, \pi_\omega, \phi)$. $\mathcal{A}_\omega$ is the *action space*

of option $\omega$, and $\mathbf{a} \in \mathcal{A}_\omega$ could be a primitive action or a different skill. $\mathcal{I} \subset \mathcal{S}$ is the *initiation set* where a skill can be started. COInS uses the common assumption that the initiation set for a skill covers the entire state space, though future work could revisit this assumption. The *skill policy* $\pi : \mathcal{S} \to P(\cdot|S = \mathbf{s})$ defines the behavior of the skill as a conditional distribution over $\mathcal{A}_\omega$. $\phi : \mathcal{S} \to [0, 1]$ is the *termination function*, which indicates the probability of a skill ending in a particular state. In this work we use deterministic terminations $\phi : \mathcal{S} \to \{0, 1\}$, which terminate if a condition (eg. interaction) is met.

This work also builds from *goal-based skills*, that parameterize the skill policy and termination function by goals $\mathbf{c}_\omega \in \mathcal{C}_\omega$. The termination function is thus a binary check of proximity to the goal $\phi(\mathbf{s}) = \|\mathbf{c}_\omega - \mathbf{s}\| < \epsilon$ and the skill policy is augmented to be $\pi : \mathcal{S} \to P(\cdot|S = \mathbf{s}, C_\omega = \mathbf{c}_\omega)$, that is, parameterized by the goal $\mathbf{c}_\omega$. *Factored goal-based skills* utilize factorization in the goals, so the goal space is a subset of a particular factor: $\mathcal{C}_\omega \subseteq \mathcal{S}_i$, and the termination function is factor proximity: $\phi(\mathbf{s}, \mathbf{c}_\omega) = \|\mathbf{c}_\omega - \mathbf{s}_i\| < \epsilon$.

A *chain of factored goal-based skills* combines levels of skills (Konidaris & Barto, 2009) in a sequence $\{\omega_0, \ldots, \omega_N\}$ where the goals of $\omega_i$ is the action space of $\omega_{i+1}$. $\mathcal{C}_{\omega_i} \subseteq \mathcal{S}$ is the space of goals used for $\phi_i(\mathbf{s}, \mathbf{c}_{\omega_i}), \pi_{\omega_i}(\mathbf{s}, \mathbf{a}, \mathbf{c}_{\omega_i})$, and is the *same* as the action space of the skill policy $\omega_{i+1}$: $\mathcal{A}_{\omega_{i+1}} = \mathcal{C}_{\omega_i}$. This structure exploits compositionality to reduce the effective horizon of upper-level skills. This work efficiently learns, unsupervised, a chain of factorized goal-based skills that can achieve high performance with good sample efficiency and transfer to similar tasks.

## 4 Chain of Interaction Skills

The Chain of Interaction Skills (COInS) algorithm is an unsupervised HRL algorithm that constructs a chain of factored goal-based skills using interactions identified using adapted Granger Causality. In this section, we first identify how interactions are identified using Granger Causality, then how the signal can be used to define a skill, and finally how goal-based RL methods can be used to learn that skill. COInS iteratively learns pairwise skills where one factor (the "source factor" with state $\mathbf{s}_a \in \mathcal{S}_a$) is controlled to produce an interaction in the "target factor" with state $\mathbf{s}_b \in \mathcal{S}_b$. $\mathbf{s}_b'$ denotes the next state of the target factor.

### 4.1 Granger Causality

The Granger causal test (Granger, 1969) is a hypothesis test to determine if the state of the source factor $\mathbf{s}_a^0, \ldots, \mathbf{s}_a^{T-1}$ is useful in predicting the target factor $\mathbf{s}_b^1, \ldots, \mathbf{s}_b^T$. Without the Markov assumption, it utilizes a history window $w$, the number of past steps needed to identify an interaction. The test compares the null hypothesis, which is the *passive* or autoregressive (self-predicting) affine model of $b$, where $\theta$ are parameters learned by affine regression to best fit $\mathbf{s}_b^{(t)}$ with noise $\epsilon_b$:

$$m^{\mathrm{pas,G}}(\mathbf{s}_b^{(t-w)}, \ldots, \mathbf{s}_b^{(t-1)}; \theta) = \theta^0 + \left[\sum_{i=1}^{w} \theta^i \mathbf{s}_b^{t-i}\right] + \epsilon_b \tag{1}$$

Then, the hypothesized relationship with signal $a$ is modeled by the *active* distribution with learned $\psi_a, \psi_b$:

$$m^{\mathrm{act,G}}(\mathbf{s}_a^{(t-w)}, \ldots, \mathbf{s}_a^{(t-1)}, \mathbf{s}_b^{(t-w)}, \ldots, \mathbf{s}_b^{(t-1)}; \psi_b, \psi_a) = \psi^0 + \left[\sum_{i=1}^{w} \psi_b^i \mathbf{s}_b^{t-i} + \psi_a^i \mathbf{s}_a^{t-i}\right] + \epsilon_a \tag{2}$$

Signal $a$ is said to Granger-cause (G-cause) signal $b$ if the regression model in Equation 2, $m^{\mathrm{act,G}}$ yields a statistically significant improvement in prediction over the autoregressive distribution in Equation 1.

### 4.2 Granger Causal Factor Tests

Granger Causality with dynamics models in FMDPs introduces two differences: First, transition dynamics in FMDPs are not always affine, so we replace the affine models with a function approximator, following recent work by Tank et al. (2021). We model $m^{\mathrm{pas,G}}, m^{\mathrm{act,G}}$ with neural conditional Gaussian models. These models use factored states as input and output the conditional mean $\mu$ and diagonal variance $\Sigma$ of a normal distribution $\mathcal{N}(\mu, \Sigma)$ over $S_b'$—ie. predicting the next target state. Second, the Markov property allows us

to collapse the history window to just the last state: $w = 1$. Combined, we describe the autoregressive and pair-regressive distributions with passive model $m^{\mathrm{pas}}$ and active model $m^{\mathrm{act}}$:

$$m^{\mathrm{pas}}(\mathbf{s}_b; \theta) : \mathcal{S}_b \to \mathcal{N}(\mu, \Sigma) \tag{3}$$

$$m^{\mathrm{act}}(\mathbf{s}_a, \mathbf{s}_b; \psi) : \mathcal{S}_a \times \mathcal{S}_b \to \mathcal{N}(\mu, \Sigma) \tag{4}$$

We call $m^{\mathrm{act}}$ the "active model" since $m^{\mathrm{act}}$ can capture when the source factor affects the dynamics of the target $P(S_b'|S_b, S_a)$. We call $m^{\mathrm{pas}}$ the "passive model" because it uses only $S_b$ to predict $S_b'$. Figure 2 illustrates the passive and active Granger models in the paddle-ball case. Using dataset $\mathcal{D}$ of state, action, next state tuples $(\mathbf{s}, \mathbf{a}_{\mathrm{prim}}, \mathbf{s}') \in \mathcal{D}$, collected during skill learning as described in Section 4.5, we train $m^{\mathrm{pas}}$ and $m^{\mathrm{act}}$ as variational models to maximize the log-likelihood of the observed data, where $m(\cdot)[\mathbf{s}_b']$ denotes the probability of next factored state $\mathbf{s}_b'$ under the modeled distribution:

$$\ell_{\mathrm{pas}}(\mathbf{s}_a, \mathbf{s}_b'; \theta) := \log m^{\mathrm{pas}}(\mathbf{s}_b; \theta)[\mathbf{s}_b'] \tag{5}$$

$$\ell_{\mathrm{act}}(\mathbf{s}_a, \mathbf{s}_b, \mathbf{s}_b'; \psi) := \log m^{\mathrm{act}}(\mathbf{s}_a, \mathbf{s}_b; \psi)[\mathbf{s}_b'] \tag{6}$$

The Markov-Granger (MG) score identifies if a source factor affects a target factor according to dataset $\mathcal{D}$:

$$\mathrm{Sc}_{MG}(D) := \left( \max_{\psi} \frac{1}{|\mathcal{D}|} \sum_{(\mathbf{s}_a, \mathbf{s}_b, \mathbf{s}_b') \in \mathcal{D}} \ell_{\mathrm{act}}(\mathbf{s}_a, \mathbf{s}_b, \mathbf{s}_b'; \psi) \right) - \left( \max_{\theta} \frac{1}{|\mathcal{D}|} \sum_{(\mathbf{s}_b, \mathbf{s}_b') \in \mathcal{D}} \ell_{\mathrm{pas}}(\mathbf{s}_b, \mathbf{s}_b'; \theta) \right) \tag{7}$$

A high score indicates that $P(S_b'|S_b, S_a) \neq P(S_b'|S_b)$, or that the source factor ($a$) generally exerts an effect on the target factor ($b$), within the context of dataset $\mathcal{D}$. However, this test identifies general relationships instead of interactions—it determines if two factors are related *overall*, but not the *specific event* when they relate—an interaction. In the next section, we utilize the active and passive models to identify interactions.

### 4.3 Detecting Interactions

Again, an MG-causal test describes whether source $a$ could be useful when predicting target $b$ in *general*, but does not detect the event where the source state $\mathbf{s}_a$ interacts with the target state $\mathbf{s}_b$ at a *particular* transition $(\mathbf{s}_a, \mathbf{s}_b, \mathbf{s}_b')$. This distinction distinguishes general causality (the former) from actual causality (the latter). Actual cause is useful in many domains where an MG-causal source and target factors interact in only a few states. For example, in Breakout the paddle's influence on the ball is MG-causal but the paddle only affects certain transitions—ball-paddle bounces. In a robot block-pushing domain, while the gripper to block is MG-causal, the gripper only interacts when it is directly pushing the block. We are often particularly interested in those particular states where the ball bounces off of the paddle, and the gripper pushes the block. In this work, "interaction" is directional from source to target.

To detect these interactions we extend the intuition of Granger-causality: an interaction is when the source factor is *specifically* useful for predicting the dynamics of the target factor. The interaction detector $h_{a,b}$ compares the active model $m^{\mathrm{act}}$ and passive model $m^{\mathrm{pas}}$ log-likelihoods in state transition $\mathbf{s}_a, \mathbf{s}_b, \mathbf{s}_b'$ to detect interactions. With $\ell$ as defined in Equations 5, 6:

$$h_{a,b}(\mathbf{s}_a, \mathbf{s}_b, \mathbf{s}_b'; \psi, \theta) := (\ell_{\mathrm{act}}(\mathbf{s}_b, \mathbf{s}_a, \mathbf{s}_b'; \psi) > \epsilon_{\mathrm{act}}) \wedge (\ell_{\mathrm{pas}}(\mathbf{s}_b, \mathbf{s}_b'; \theta) < \epsilon_{\mathrm{pas}}) \tag{8}$$

Here, $\epsilon_{\mathrm{act}}$ and $\epsilon_{\mathrm{pas}}$ are hyperparameters based on the environment's inherent stochasticity; additional discussion is in Appendix I. The first condition ensures $m^{\mathrm{act}}$ predicts the next state with sufficiently high accuracy (log-likelihood higher than $\epsilon_{\mathrm{act}}$), and the second ensures that $m^{\mathrm{pas}}$ predicts with low accuracy (log-likelihood lower than $\epsilon_{\mathrm{pas}}$). This infers interactions using the following logic: states with low passive log-likelihood $\ell_{\mathrm{pas}}(\mathbf{s}_b, \mathbf{s}_b') < \epsilon_{\mathrm{pas}}$ could arise from three cases: (1) when state transitions are highly stochastic, or (2) some other factor $\neq a$ is interacting with $b$ or (3) the source factor interacts with the target factor. In cases (1) and (2), $\ell(\mathbf{s}_b, \mathbf{s}_a, \mathbf{s}_b'; \psi)$ will be low. Figure 2 includes qualitative illustration in Breakout.

When searching for MG-causal relationships, the MG score (Equation 7) can often be small when interactions are extremely rare. In Breakout, paddle-ball interactions only occur once every $\sim$1000 time steps

Figure 2: **Left:** An illustration of the active and passive model inputs and outputs for the paddle-ball Granger models. **Right:** Three possible-interaction states. In Case 1, COInS predicts no interaction because both the passive and active models predict accurately. In Case 2, the active model predicts accurately using paddle information, but the passive model does not, indicating a paddle-ball interaction. Case 3 is not a paddle-ball interaction since *both* the passive and active models predict poorly.

when taking random actions, so summary statistics can fail to efficiently capture this relationship. Similarly, gripper-block interactions in Robot Pushing occur every ∼1500 time steps. To account for this, we adjust the MG-score by modulating it with the interactions, giving the interaction score $\text{Sc}_I$. For $N_{\text{int}} = \sum_{\mathbf{s}_a, \mathbf{s}_b, \mathbf{s}'_b \in \mathcal{D}} h_{a,b}(\mathbf{s}_a, \mathbf{s}_b, \mathbf{s}'_b; \psi, \theta)$ as the number of detected interactions in $\mathcal{D}$:

$$\text{Sc}_I(D, a, b) := \frac{1}{N_{\text{int}}} \sum_{\mathbf{s}_a, \mathbf{s}_b, \mathbf{s}'_b \in \mathcal{D}} h_{a,b}(\mathbf{s}_a, \mathbf{s}_b, \mathbf{s}'_b; \psi, \sigma) \left(\ell_{\text{act}}(\mathbf{s}_a, \mathbf{s}_b, \mathbf{s}'_b; \psi) - \ell_{\text{pas}}(\mathbf{s}_b, \mathbf{s}'_b; \theta)\right) \tag{9}$$

### 4.4 Using Interactions in HRL

Now that we have the means to detect interactions between two factors, our next step is to train a new skill that leverages existing source factor skills to learn to control a target factor. This requires three design choices: selecting the source factor, identifying the target factor, and learning control over the target factor. For the first choice, COInS uses the most recently acquired skill's corresponding factor as the source factor, creating a chain, but future work can extend this to tree or DAG hierarchies.

#### 4.4.1 Determining Target Factor Control

COInS selects the target factor for learning by evaluating $\text{Sc}_I(\mathcal{D}, a, \hat{b})$ for all uncontrolled $\hat{b}$. $a$ indexes the most recently controllable factor, and $\hat{b}$ indexes a factor yet to be controlled by COInS. The target factor $b$ is then chosen based on the highest interaction score, as long as it is higher than a minimum threshold $\epsilon_{\text{SI}}$:

$$b = \arg\min_{\hat{b}} \text{Sc}_I(\mathcal{D}, a, \hat{b}) \quad \text{s.t.} \quad \text{Sc}_I(\mathcal{D}, a, \hat{b}) > \epsilon_{\text{SI}} \tag{10}$$

However, just because one *factor* ($a$) can affect the dynamics of another ($b$) does not mean that every *feature* of $\mathbf{s}_b$ is controllable. Certain features $\mathbf{s}_b[k]$ of the target factor $\mathbf{s}_b$ might not be directly controllable. For example in Breakout, the paddle affects the ball dynamics, but within one timestep of an interaction, the ball position cannot be directly controlled. Trying to control both the ball position and velocity simultaneously would result in many infeasible goals. To address this, COInS alters the goals by determining a control mask $\eta_b \in \{0, 1\}^K$ where $K$ is the number of features in factor $\mathbf{s}_b$.

Controllable features are identified based on this intuition: if the feature value differs significantly from the passive prediction after an interaction with the source factor when evaluated over the dataset $D$, the source factor likely caused the change. $\mu(m^{\text{pas}}(\mathbf{s}_b; \theta))$ denotes the mean of the passive prediction and $N_{\text{int}}$ as defined in Section 4.3, the masks ($\eta_b$) are defined with:

$$\eta_b[k] = \begin{cases} 1 & \frac{1}{N_{\text{int}}} \sum_{\mathbf{s}_a, \mathbf{s}_b, \mathbf{s}'_b \sim \mathcal{D}} h_{a,b}(\mathbf{s}_a, \mathbf{s}_b, \mathbf{s}'_b; \psi, \theta) \left\| \mathbf{s}'_b[k] - \mu(m^{\text{pas}}(\mathbf{s}_b; \theta)) \right\|_1 > \epsilon_\eta \\ 0 & \text{otherwise,} \end{cases} \tag{11}$$

For examples of $\eta_b$-masks and post-interaction states in evaluated domains, see Section 5. In an abuse of notation, we define $\mathcal{C}_b := \eta_b \mathcal{S}_b$ to be the goal space for our options, even though it is a space of masked target states $\mathbf{s}_b$. If the size of the unique masked post-interaction state set $\mathcal{C}_b := \text{unique}(\eta_b \mathcal{S}_b)$ is sufficiently small ($|\mathcal{C}_b| < n_{\text{disc}}$) then the goal space is treated as discrete, otherwise, it is treated as continuous.

### 4.4.2 Training Factor Control Skills

With the target factor and space of goals over that target factor determined, COInS can now perform goal-based reinforcement learning with factored interaction goals. A specific goal from the masked target factor space $\mathbf{c}_b \subseteq \mathcal{C}_b$ parameterizes the skill. The termination function of the skill's temporal extension is represented with a binary indicator function. This function indicates when the goal is reached $\eta_b \mathbf{s}_b = \mathbf{c}_b$ co-occurring with a detected interaction $h_{a,b}(\mathbf{s}_a, \mathbf{s}_b, \mathbf{s}_b'; \psi, \theta)$. With mask $\eta_b$, source and target states $\mathbf{s}_a, \mathbf{s}_b, \mathbf{s}_b'$ and goal target features $\mathbf{c}_b$:

$$\phi_b(\mathbf{s}_a, \mathbf{s}_b, \mathbf{s}_b', \mathbf{c}_b) := \begin{cases} 1 & h_{a,b}(\mathbf{s}_a, \mathbf{s}_b, \mathbf{s}_b'; \psi, \theta]) \wedge \|\eta_b \mathbf{s}_b' - \mathbf{c}_b\|_1 < \epsilon_c \\ 0 & \text{otherwise.} \end{cases} \tag{12}$$

The associated reward function is $R(\mathbf{s}, \mathbf{a}, \mathbf{s}') := \phi_b(\mathbf{s}_a, \mathbf{s}_b, \mathbf{s}_b', \mathbf{c}_b) - \epsilon_{\text{rew}}$, *i.e.* 0 at the goal and $-\epsilon_{\text{rew}}$ elsewhere. By applying goal-based RL to this reward, an iteration of COInS learns to produce a particular pairwise interaction (ex. ball bounce), that achieves a goal (ex. particular ball angle). The skill uses the goals of the last learned skill as a temporally extended action space.

The goal-based RL used to train our goal-reaching skills utilizes hindsight experience replay (Andrychowicz et al., 2017), with Rainbow (Hessel et al., 2018) for discrete action spaces, and Soft Actor-Critic (Haarnoja et al., 2018b) for continuous action spaces. COInS uses only $\mathbf{s}_a, \mathbf{s}_b$ instead of all of $\mathbf{s}$ as input to the policy to accelerate learning by having the agent attend only to the minimum necessary aspects of the state. A skill is considered learned when the goal-reaching reward remains unchanged after $N_{\text{complete}}$ training steps.

### 4.5 Building the Skill Chain

The previous sections detail the procedure for learning a *single* link in the chain of skills. COInS iteratively adds links one skill at a time until no new links are found through Equation 10. COInS starts with primitive actions as the only source factor and takes random actions. Each iteration: 1) learns active and passive models for each uncontrolled factor where the source is the last factor COInS learned control over. 2) Finds the next target factor using Equation 10. 3) Learns a goal-based policy with the reward function described in Section 4.4.2 based on Equation 12. 4) Adds factor $b$ to the chain and restarts the loop.

Each step of the loop is executed autonomously, and the data used to evaluate the interaction test is collected from the RL training process and thus self-generated. The loop ends when there are no more high-score factors to control: $\text{Sc}_I < \epsilon_{\text{SI}}$, where $\epsilon_{\text{SI}}$ is a hyperparameter. Algorithm box 9 describes the algorithm. The benefits of COInS come from three sources: 1) COInS breaks down complex environments into a series of transferable, intuitive skills automatically. 2) each skill is individually simpler than the overall task, resulting in improved sample efficiency. 3) By incorporating the causal test for interactions, the skills exploit sparse dynamic relationships for control.

---

**Input:** FMDP Environment $E$
**Initialize** Chain of Interaction Skills $\vec{\omega}$ with a single option set $\omega_{\text{prim}}$. Assign $a = $ primitive actions
**Data** $\mathcal{D} = \mathcal{D} \bigcup \text{random policy data}$
**repeat**
  **Interaction Detector**: Optimize likelihoods (Equations 5, 6) on $\mathcal{D}$ to get the passive and active models $m^{\text{pas}}, m^{\text{act}}$ and interaction detector $h_{a,b}$ (Equation 8) for all uncontrolled $b$.
  **Interaction Test**: Find the candidate target object with Equation 10 or terminate ($\max_b \text{Sc}_I < \epsilon_{\text{SI}}$)
  **Option Learning**: Determine $\eta_b$ (Equation 11) and learn interaction skills using reward $\phi_b - \epsilon_{\text{rew}}$
  **Update** Add $\omega_b$ to $\vec{\omega}$. Progress $a = b$. Append dataset $\mathcal{D} = \mathcal{D} \bigcup \text{option learning data}$
**until** COInS no longer finds any interaction test pairs

---

# 5 Experiments

We systematically evaluate COInS in two domains: 1) an adapted version of the common Atari baseline Breakout (Bellemare et al., 2013) (Figure 1 and Appendix A.1) and 2) a simulated Robot pushing domain in robosuite (Zhu et al., 2020) with randomly generated negative reward regions (Figure 1 and Appendix A.2). In these domains, we compare COInS against several baselines: a non-hierarchical baseline that attempts to learn each task from scratch, a fine-tuning-based transfer with a 5M step pre-trained neural model, an adapted version of Hindsight Actor Critic (HAC) (Levy et al., 2019b), a model based causal RL algorithm Causal Dynamics Learning (CDL) (Wang et al., 2022), a curiosity and count-based exploration method, Rewarding Impact-Driven Exploration (RIDE) (Raileanu & Rocktäschel, 2020) and a hierarchical option chain algorithm Hypothesis Proposal and Evaluation (HyPE) (Chuck et al., 2020). Our results show that COInS is more sample efficient, achieves higher overall performance, and also learns skills that generalize well to a variety of in-domain tasks. COInS' learning procedure does not require human intervention—it progresses automatically based on the performance of the interaction score and the goal-reaching policies.

Details about the skill learning procedure, including the number of time steps used for learning each level of the hierarchical chain, the interaction masks, and interaction rates are found in Appendix G for Breakout, Appendix H for Robot pushing and Table 4. Hyperparameter discussion is also in Appendix I. In this section, we discuss the performance comparison with baselines.

We represent all the policies, including the baselines, with an order-invariant PointNet Qi et al. (2017) style architecture and use factored states as inputs. When there are only two factors, this is similar to a multi-layer perceptron. Details about architectures are found in Appendix F.

## 5.1 Overview of Baselines

Before describing the empirical results, we briefly detail how the baselines compare with the Granger-causal and hierarchical elements of COInS. Even though prior work has applied RL to Breakout and robotic manipulation, not all baselines have been applied to these domains. We modify and tune the baselines to improve their performance. The Breakout variants and Robot pushing with negative reward regions are novel tasks to demonstrate how reward-free Granger-causal skill discovery can improve RL performance.

**HAC** (Levy et al., 2019b) is a reward-based HRL method that uses a goal-reaching chain of skills, where hierarchical levels each have a horizon of $H$. This assesses COInS against a multilevel reward-based hierarchical method to illustrate how interaction-guided skills compare against a hierarchical method without interactions. Default HAC is infeasible in both the Breakout and Robot pushing tasks because of the high dimensionality of the state spaces. To improve performance, we introduce domain knowledge through hand-chosen state factors as goal spaces. In Breakout, the lower-level skills use the ball velocity as goals. In Robot pushing, they use the block position. Additional skill hierarchy results are in Appendix J.

**CDL** (Wang et al., 2022) is a state-of-the-art causal method that learns causal edges in the factored dynamics for model-based reinforcement learning. It constructs a general causal graph to represent the counterfactual dynamics between state factors. Without identifying interactions, CDL struggles in these domains because CDL fails to capture infrequent interactions even when it picks up general relationships. In addition, it must learn models over all the objects, of which there are many. CDL illustrates how general causal reasoning without interaction can struggle in rare-interaction domains with many factors.

**RIDE** (Raileanu & Rocktäschel, 2020): Rewarding impact-driven exploration (RIDE) is an intrinsic reward method that combines curiosity with neural state counts to perform efficient exploration. This method disambiguates COInS skill learning from state-covering exploration bonuses and the advantage of directed behavior through interactions. While RIDE has been applied successfully to visual agent maze navigation settings, these domains often lack the combinatorial complexity of Breakout or Robot pushing. Our implementation of RIDE utilizes the Pointnet-based architectures for the forward model with neural hashing, with Rainbow and SAC for RL. Additional details and discussion of RIDE can be found in Appendix J.4.

**HyPE** (Chuck et al., 2020) constructs a skill chain reward-free through interactions and causal reasoning, akin to COInS. Unlike COInS, which learns goal-oriented skills, HyPE uses clustering to acquire a discrete skill set, guided by physical heuristics such as proximity and quasi-static dynamics. Despite these differ-

|       | Vanilla (5M) | HAC (5M) | R-HyPE (5M) | CDL (5M) | RIDE (5M) | COInS (<0.7M) |
|-------|--------------|----------|-------------|----------|-----------|---------------|
| Break | $85.6 \pm 8.3$ | $74.3 \pm 8.2$ | $\mathbf{95.3 \pm 3}$ | $-50 \pm 1.7$ | $\mathbf{92.5 \pm 8}$ | $90.7 \pm 15$ |
| Push  | $-30 \pm 0.01$ | $-43 \pm 24$ | $-90 \pm 19$ | $-30 \pm -0.1$ | $-31 \pm 2$ | $\mathbf{-21.6 \pm 4.6}$ |

Table 1: **Sample efficiency** for Breakout (Break) and Robot Pushing (Push) with negative reward regions. Note that $-30$ in Robot Pushing (vanilla, CDL) reflects the performance of a policy that never touches the block. Baselines are evaluated after 5M time steps of training, while COInS gets equivalent or better performance in 300k/700k timesteps for Breakout/Robot Pushing, respectively. The final performance after COInS is trained for 1-2M time steps is shown in Table 5 in the Appendix.

ences, HyPE is the closest comparison for COInS since it learns a skill chain based on interactions. HyPE performance illustrates how the Granger-causal interaction models and goal-based hierarchies allow more powerful and expressive skill learning. HyPE is designed for pixel spaces, but we adapted it to use the true factor states. We evaluate two variants of HyPE: R-HyPE uses the RL algorithm Rainbow (Hessel et al., 2018) to learn policies, while C-HyPE uses CMA-ES (Hansen et al., 2003). When evaluating C-HyPE sample efficiency, we add together the cost of policy evaluation for the policies and graph the performance of the best policy after each update, following the methods used in HyPE.

**Vanilla RL** uses Rainbow (Hessel et al., 2018) for discrete action spaces and soft actor-critic (Haarnoja et al., 2018b) for continuous action spaces. These baselines are chosen as the most stable RL methods for achieving high rewards on new tasks. In Breakout variants, we evaluate transfer using a pre-train and **fine-tune** strategy by pretraining on the source task and then fine-tuning on the variant task.

We chose not to compare with diversity-based skill learning methods for Breakout and Robot pushing due to the considerable challenge these methods face in such high-dimensional spaces—these methods are typically applied to low-dimensional locomotion or navigation. Breakout and Robot Pushing both have state spaces that are quite large: Breakout has 104 objects, and Robot Pushing has 18. Reaching a large number of these states is often infeasible. Designing a suitable diversity-based method for this would require a significant reengineering of existing methods.

## 5.2 Sample Efficiency

In this section, we discuss how COInS achieves improved sample efficiency by learning interaction-controlling skills. Specific learning details can be found in Appendix A.1 and A.2, and training curves in Figure 3.

In Breakout, COInS learns high performance 4× faster than most of the baselines. This comes from utilizing the goal-based paddle control to shorten the time horizon of the ball bouncing task, and shortening credit assignment between bounces by the duration of paddle goal-reaching. COInS achieves high rewards in Breakout without optimizing extrinsic reward since good performance only requires bouncing the ball. COInS skills exceed this by learning to control the ball to desired bounce angles.

HAC can learn only after the changes in Section 5.1. Even then, it performs poorly—worse than the vanilla RL baseline. We hypothesize this is because it is difficult to apply credit assignments through a hierarchy effectively. HyPE performs comparably to COInS in sample efficiency. C-HyPE even outperforms COInS, because the evolutionary search can very quickly identify the paddle-ball relationship—which supports the hypothesis that interactions are a good guide for agent behavior. However, while C-HyPE can control bounces, it struggles to identify and control bounce *angles*, resulting in poor transfer (Figure 4).

The sample efficiency of COInS highlights the advantage offered by interaction-guided skill acquisition, allowing the breakdown of complex behaviors using interaction signals and reducing the time horizon through skill learning. The comparatively similar performance of HyPE supports the efficacy of interactions since HyPE also uses interaction signals to achieve high performance. COInS and other interaction-based methods are likely to perform well in domains where sparse interactions have a significant impact on performance, such as domains with sparse contact or large numbers of distant objects.

## 5.3 Overall Performance

Many of the baselines can achieve near-optimal performance in Breakout (Table 1). The same is not true in Robot block pushing with negative reward regions task (see Appendix A.2 for details). In this task, no

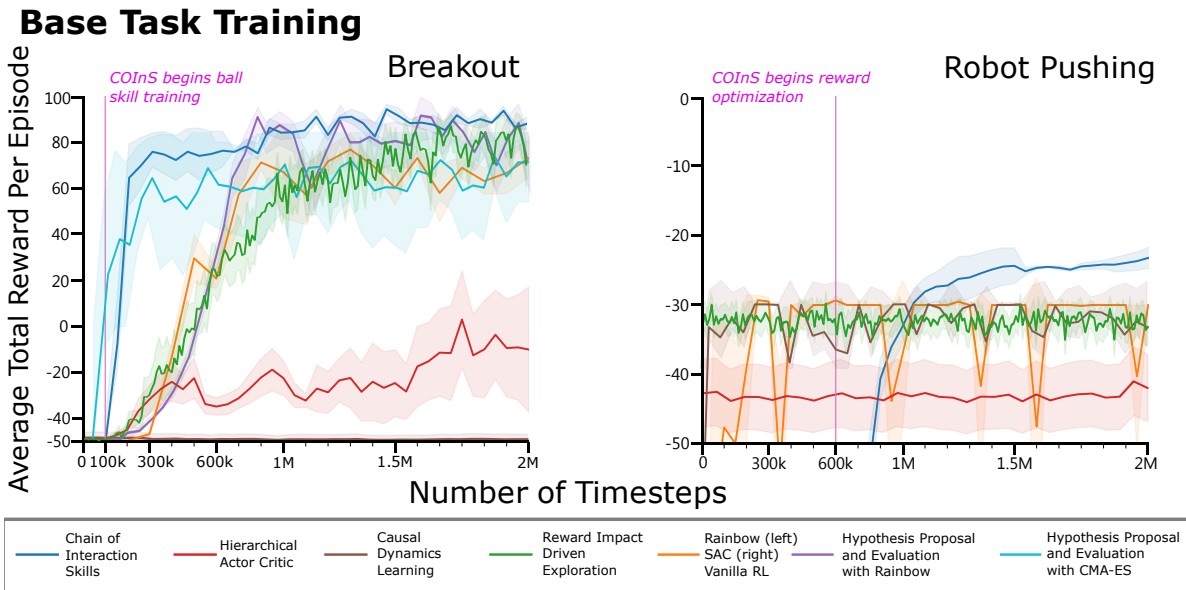

Figure 3: Each algorithm is evaluated over 10 trials with the shaded region representing standard deviation. Training performance of COInS (blue) against baselines (see legend) on **Breakout (left)** and negative reward regions **Robot Pushing (right)**. The vertical pink lines indicate when COInS starts learning a high-reward policy. In the pushing domain the return for not moving the block is $-30$. Most algorithms do not touch the block. The minimum total reward is $-600$—by spending every time step in a negative region. with R-HyPE's performance falls within this lowest bracket. Final evaluation of COInS against the baselines after $2M$ steps is found in Appendix Table 5.

| Algo | single | hard | big | neg | center | prox |
|---|---|---|---|---|---|---|
| Vanilla | $-5.8 \pm 0.5$ | $-7.1 \pm 0.8$ | $\mathbf{0.74 \pm 0.1}$ | $\mathbf{2.9 \pm 0.3}$ | $-37 \pm 11$ | $0.1 \pm 0.2$ |
| Fine-Tuned | $-5.4 \pm 0.9$ | $-6.1 \pm 0.4$ | $\mathbf{0.79 \pm 0.1}$ | $-7.0 \pm 19$ | $-42 \pm 9$ | $0.2 \pm 0.1$ |
| R-HyPE | $-5.6 \pm 0.45$ | $-5.1 \pm 0.49$ | $0.67 \pm 0.06$ | $\mathbf{2.9 \pm 0.46}$ | $-20 \pm 1.5$ | $-0.3 \pm 0.3$ |
| COInS | $\mathbf{-3.2 \pm 1.2}$ | $\mathbf{-4.2 \pm 0.9}$ | $\mathbf{0.85 \pm 0.06}$ | $\mathbf{3.6 \pm 0.3}$ | $\mathbf{-12 \pm 4}$ | $\mathbf{0.5 \pm 0.05}$ |

Table 2: **Transfer evaluation of trained policies on Breakout variants**. "Fine-tuned" fine-tunes a model pre-trained on the base task, and "Vanilla" trains from scratch. The baselines are evaluated after 5M time steps, while COInS achieves superior performance in 500k (hard, neg) and 2M (center). Figure 4 contains descriptions of the variants, and additional baseline transfer results are in Appendix Table 5

method, even COInS, achieves perfect performance. This is because the tabletop is densely populated with negative reward regions in highly variable configurations between episodes, (see Figure 1). The task is also long horizon, taking as many as 300-time steps to complete, and reward shaping is difficult because pushing the block to the goal must be mediated by the negative regions.

COInS discovers block control through the interaction test and optimizes reward using block movements as actions. This results in a complex policy that pushes the block between the negative regions. None of the baselines can achieve this policy: CDL learned dynamics models fail to capture the rare dynamics of the gripper-block interaction. HAC and Vanilla RL use rewards, which trap the learned policies in the local minima of never touching the block. While HyPE policies can learn to move the block, they lack fine-grained control when pushing the block to desired locations. See Figure 3 for details.

COInS block pushing results demonstrate how interaction-controlling skills reframe a challenging temporally extended task into one that is feasible for RL. Without using interactions, these skills are difficult to discover from reward, as seen with policies learned with HAC. The failure of exploration methods demonstrates that without interactions, state-covering struggles as well. Interaction-based methods offer a tool for directed pretraining of factorized control before optimizing reward.

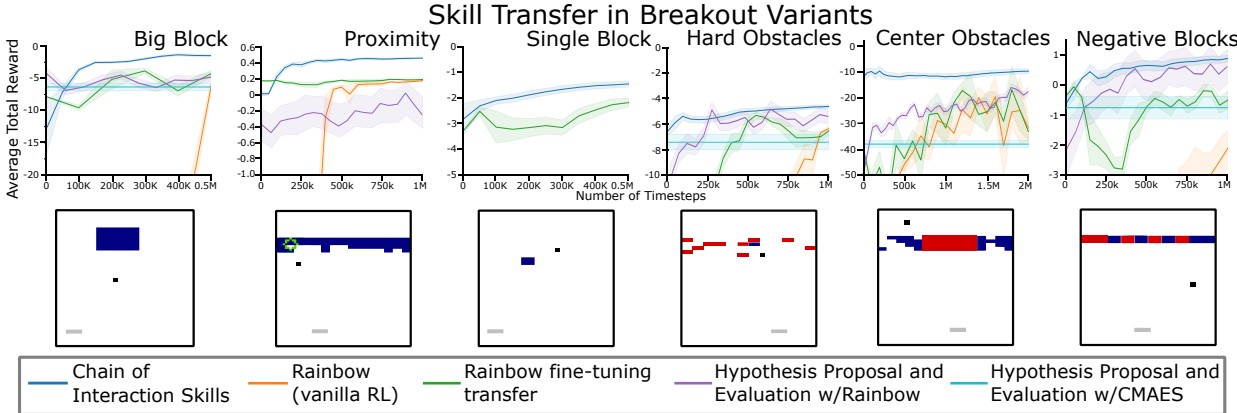

Figure 4: **Skill transfer** with COInS (blue), training from scratch (orange), pre-train and fine-tune to the variant (green), and HyPE skills (C-HyPE in cyan, R-HyPE in purple). C-HyPE has nothing to optimize on transfer (it only learned a single "bounce" action), so average performance is visualized. Below the performance curves, the variants are visualized with the red (light) blocks as negative reward/unbreakable blocks, and the blue (dark) positive reward blocks. Details in Appendix J and Section 5.4.

## 5.4 Transfer

We evaluate skill transfer in variants of Breakout. These variants have complex reward structures that make them hard to learn from scratch—negative reward for bouncing the ball off the paddle, or for hitting certain blocks. These penalties make credit assignments difficult in a long horizon task like Breakout.

- **Single block (single)**: A domain with a single, randomly located block, where the agent receives a $-1$ penalty for bouncing the ball, and the episode ends when the block is hit.
- **Hard obstacles block (hard)**: The single block domain, but with 10 randomly initialized unbreakable obstacle blocks.
- **Big block (big)**: A domain where a very large block can spawn in one of 4 locations, and the agent must hit the block in one bounce or incur a $-10$ reward.
- **Negative blocks (neg)**: There are ten blocks at the top in a row. Half are randomly assigned to $+1$ reward, the other half $-1$ reward, and one episode is 5 block hits.
- **Center obstacles (center)**: The agent is penalized (-1) for bouncing the ball, and the center blocks are unbreakable, forcing the agent to learn to target the sides for reward.
- **Proximity (prox)**: a domain with a randomly selected target block, with reward scaling between 1 and $-1$ based on how close the ball is to hitting that particular block.

COInS learned skills that bounce the ball at particular angles, which transfer to learning these difficult Breakout variants. By reasoning in the space of ball bounces, policies trained in the variants can exploit temporal abstraction and simplified credit assignment to handle difficult reward structures.

CDL and HAC perform poorly on the original task, and it is unclear how to transfer their learned components. RIDE learned the same kind of policy as vanilla RL. Thus we do not evaluate transfer using these baselines. The fine-tuning strategy for transfer did show some success, as it transfers bouncing behavior to these new domains. However, it rarely learns past this initial behavior, and performance can even slightly decline. R-HyPE can transfer ball angle bouncing skills, but since the skills have many failure modes bouncing the ball, the overall performance on the variants is poor. C-HyPE does not learn to bounce the balls at different angles (it only learns a single discrete option for ball control), so it only has one action to take, and cannot transfer. It shows the baseline performance of a ball-bouncing policy on the different variants.

The results in Figure 4 demonstrate how the skills learned with COInS are agnostic to factors that are not part of the skill chain, i.e. the blocks. Like with robot pushing, using these skills can exceed the performance of vanilla RL trained directly because they reduce the time horizon for complex reward assignments. In settings where at least some of the same objects and dynamics are present, interaction-based skills can rapidly transfer and achieve superior final performance to training from scratch, especially on difficult reward tasks.

# 6 Conclusion

This work introduces a novel approach to HRL in factored environments. The key idea is to use adapted Granger-causal interaction detectors to build a hierarchy of factor-controlling skills. The COInS algorithm presents one practical method that performs well on a robotic pushing domain with negative reward regions and variants of the video game Breakout. In these domains, it shows improvement in sample efficiency, overall performance, and transfer to difficult tasks that cannot be trained from scratch with Vanilla RL. Future work can explore the current limitations of COInS, particularly 1) its reliance on the core assumptions of a single chain of skills being sufficient to represent the task, 2) pairwise interactions capturing the object dynamics, and 3) a given state factorization.

To extend COInS beyond chain hierarchies, a DAG structure of skills can be learned by selecting from any of the controllable factors and addressing the design question of multiple-factor skills as inputs. To scale to numerous interactions, future work can investigate multiple-interaction extensions of Granger Causality, such as with gradient-based measures (Hu et al., 2023). Taking this further, interactions could be differentiated between different kinds of behavior *within* a pair of objects, such as differentiating pushing, and picking. Finally, vision models such as (Kirillov et al., 2023) can address factorization and correspondence. Despite these challenges, COInS represents a significant initial step in leveraging interactions among factors, opening avenues for future work utilizing human demonstrations to handle tasks like grasping, simplifying hyperparameter complexity, tuned stopping conditions, and reducing computational cost. Overall, COInS shows evidence that controlling interactions in state factors is a promising direction for skill discovery.

# 7 Acknowledgements

This work has taken place in part in the Safe, Correct, and Aligned Learning and Robotics Lab (SCALAR) at The University of Massachusetts Amherst and the University of Texas at Austin. SCALAR research is supported in part by the NSF (IIS-2323384), AFOSR (FA9550-20-1-0077), ARO (78372-CS), and the Center for AI Safety (CAIS). The work was supported by the National Defense Science & Engineering Graduate (NDSEG) Fellowship sponsored by the Air Force Office of Science and Research (AFOSR). Special thanks to my collaborators Stephen Guigere, Yuchen Cui, Akanksha Saran, Wonjoon Goo, Daniel Brown, Prasoon Goyal, Harshit Sikchi, Christina Yuan, and Ajinkya Jain for their fruitful conversations and timely help.

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

## A   Environments

### A.1   Breakout

The Breakout domain is not the same as Atari Breakout. The Breakout domain contains as factors the actions, ball, paddle, and blocks, shown in Figure 1. It contains 100 blocks, and the agent receives +1 reward whenever it strikes a block. An end-of-episode signal and -10 reward are given whenever the agent lets the ball fall past the paddle. All other rewards for the task are 0. The ball moves at $\pm 1$ x velocity and $\pm 1, \pm 2$ y velocity, depending on how it is struck by the block. Like in Atari Breakout, the ball angle is determined by where on the paddle the ball strikes, with four different angles.

We use an adapted Breakout environment for three reasons. First, we want the domain to have stationary dynamics so that learning dynamics models is relevant. However, the Atari Breakout domain has the velocity of the ball change based on the number of bounces the agent has taken, where the number of bounces is a hidden variable. Rather than overcomplicate the learning procedure, we opted to use a domain where this feature was absent. Second, to investigate long-horizon tasks we wanted the domain to have sufficiently long episodes, which we accomplished by slowing the speed of the ball. Long horizons are often mitigated in RL algorithms through frame skipping to improve efficiency, but we are interested in investigating the granular features of the Environment like the transition dynamics. Third, since we are working in a factored space we wanted a domain where we could have easy access to the factors and related statistics like which objects are interacting with which others at a given time step. This is easier to accomplish by having a custom version rather than trying to extract that information from the RAM state of Atari Breakout.

### A.2   Robot Pushing

The Robot pushing domain uses as factors the actions, gripper, block, and negative reward "obstacles". The gripper is a 7-DOF simulated Panda arm generated in robosuite (Zhu et al., 2020) with 3-DOF position control using Operational Space Control (OSC). The OSC controller generates joint torques to achieve the desired displacement of the end effector specified by the action. The objective of the domain is to push a $5\text{cm}^3$ block to a 5cm diameter circular goal location randomly placed in a 30cm×30cm spawn area. The area is broken into a 5×5 grid, where 15 of the 25 6cm×6cm grids are negative reward regions (NRR), which implies almost $300M$ possible configurations of the obstacles, block, and target, forcing the agent to learn a highly generalizable policy. If the block enters an NRR or leaves the spawn area, the agent receives a $-2$ reward. The negative regions are generated such that there will always exist an NRR-free path to the goal. The domain also has a constant $-0.1$ reward per timestep for goal reaching, with episodes of 300 time steps. An image of the domain is shown in Figure 1.

The negative reward region locations are generated such that there will always be a trajectory to the goal location, so the actual number of possible configurations is limited by this. However, this is still more than enough complexity for most problems, and in fact, this could be framed as a generalization in the RL problem, since it is entirely possible in millions of timesteps that the agent has never seen the provided configuration. In other words, the agent must learn to reason about the negative reward regions, something that is only possible with current RL algorithms if it reasons from the space of the block positions, and not from the gripper, hence our choice.

We would have liked to use a pick-and-place task, but these tasks become infeasibly difficult to perform from purely random actions, often requiring clever hacking of the data to get the gripper to grasp the block. This would undermine the causal nature of the tests since now the actions are chosen without being agnostic to the objects. However, we are looking for adaptations

## B   Details on Active and Passive Model Training for Interactions

The interaction model often requires data balancing to train—meaning weighting the frequency of interaction to non-interaction states. For one thing, the policy used to gather data is agnostic to the interactions since if it was not this can induce a correlation between objects that come from the policy, not the dynamics. Thus,

the passive model is trained on states where the actions are taken from a policy agnostic to $b$, the target factor.

In practice, this is often sufficient to train the passive model for good prediction on states where interactions do not occur, though we have found that using a heuristic such as proximity to determine which states are unlikely to have interactions can greatly smoothen the learning process. This is because the model can often fixate on the states that it cannot predict, resulting in misprediction at states where it could otherwise perform well. This also helps to widen the gap between the passive and active model log-likelihoods, which makes $I$ more accurate

While perfect data and modeling give an $I$ that captures most interactions, interactions can vary from being extremely rare (the ball bouncing off of a randomly moving paddle), or extremely common (the actions affecting the paddle everywhere). In practice, this makes data balancing an issue when training the active model $g(\mathbf{s}_a, \mathbf{s}_b)$, where a network will end up struggling to predict the states where $\mathbf{s}_a$ is useful because it is overwhelmed by the volume of passive states. Simply adding model complexity can cause the active model to memorize states that might be hard to predict or involve an interaction with a different factor $c$, resulting in spurious interactions.

To combat this, COInS uses the following strategies: first, up-sample states with high passive error (low log-likelihood) when training the active model. This causes the active model to favor predicting states where the passive model is already performing poorly. Second, downweight states with low $a - b$ proximity to prevent the active model from overfitting to all states with high passive error, including ones where $a$ does not interact with $b$. This keeps the active model from memorizing any state. Future work could consider other, general strategies for controlling data imbalances.

These two data balancing methods can be combined in an unnormalized weight $w$ for any state that has the following passive error and proximity, with hyperparameter $\lambda$:

$$w_b = \begin{cases} \lambda + 1 & -\log m^{\mathrm{pas}}(\mathbf{s}_b; \theta)[\mathbf{s}_b'] > 0 \wedge \|\mathbf{s}_b - \mathbf{s}_a\| < \epsilon_{\mathrm{close}} \\ 1 & \text{otherwise} \end{cases} \tag{13}$$

We also propose a possible way to tune the active model with interactions, by weighting the loss of the active model by the interaction model. This has to be balanced from overfitting, however, as this can make the interaction model boost the active model. This tuning would make the forward loss with weighted dataset $\mathcal{D}_w^b$:

$$\min_{\phi} L(m^{\mathrm{act}}, \mathcal{D}_w^b) := E_{\mathbf{s}_a, \mathbf{s}_b, \mathbf{s}_b' \sim \mathcal{D}_w^b}[\lambda_i \log \left( m^{\mathrm{act}}(\mathbf{s}_a, \mathbf{s}_b; \phi)[\mathbf{s}_b'] \right)$$
$$+ (1 - \lambda_i) I(\mathbf{s}_a, \mathbf{s}_b, \mathbf{s}_b') \log \left( m^{\mathrm{act}}(\mathbf{s}_a, \mathbf{s}_b; \phi)[\mathbf{s}_b'] \right)] \tag{14}$$

Where $\lambda_i$ is the mixing parameter. However, combined with the proximity this can end up being too much boosting and result in overfitting in the active model.

We also tried using a learned interaction model instead of a decision test: $I(\mathbf{s}_a, \mathbf{s}_b; \theta) : \mathcal{S}_a \times \mathcal{S}_b \to [0, 1]$, the probability of an interaction at a given state, and learns to predict the interaction detections $I(\mathbf{s}_a, \mathbf{s}_b, \mathbf{s}_b')$. This could have the benefit of generalizing the characteristics of an interaction to states where $\mathbf{s}_a'$ might be hard to predict. However, because we continuously train the active model with new data from the policy, this turns out not to perform well because it complicates the learning process.

## C  Object Centric Actions and State Augmentations

When using the goal space of the current option as actions, there are two important design questions: should the action space be continuous or discrete, and should the actions be in a fixed space, or relative to the state of the target object?

In general goal-based RL both discrete and continuous action spaces are possible, but when the goal space is small, such as the ball velocity, discrete spaces are preferable. This is because the space of states seen

after an interaction $\mathcal{C}_{b'}$, after being further reduced by the controllable elements mask $\eta_b$, can be quite small. When this space is small, the locality that is present in normal state spaces might not be present, so selecting continuously could add bias to "nearby" states. Thus, if $\mathcal{C}_{b'} \cdot \eta_b <= 10$, or there are less than 10 seen interaction states, then we use discrete actions.

For the second question, continuous actions are generally better when relative to the agent/object being controlled. This is because continuous spaces often have a sense of locality—eg. the paddle in Breakout is close to states with close values. We encode relative state in the action space by having policy actions $a_\pi$ between -1 and 1, and then remapping that into a relative factor space, or $a_\pi \cdot d \cdot \eta + \mathbf{s}_b$, where $d$ is the relative distance a single action can go, which is typically 0.2 of the normalized space $\mathcal{S}^b$.

## D Causality and Object Interactions

While this work introduces a novel metric for identifying entity interactions, similar ideas in mechanisms like learning causal relationships (Pearl, 2009) have been incorporated in model-based and planning-based methods to explain dynamics (Li et al., 2020; Wang et al., 2022). Schema networks (Kansky et al., 2017) and Action Schema Networks (Toyer et al., 2018) extend causal models to planning. Alternatively, variational methods such as (Lin et al., 2020) learn to identify objects in a visual scene for planning (Kossen et al., 2020), or reinforcement learning (Veerapaneni et al., 2020). By contrast, COInS limits the modeling burden by only capturing forward dynamics models good enough to identify interactions and using them for HRL, drawing ideas from empowerment (Jung et al., 2011) and contingency (Bellemare et al., 2012) to improve exploration. The model-disagreement technique used by COInS builds on neural methods for Granger Causality (Granger, 1969; Tank et al., 2021), though this is a novel application to HRL.

In this work we differ from general causality in two ways: first, we acknowledge that the Granger-causal test does not demonstrate causality, but rather is a predictive hypothesis test designed for time series. However, in the case of forward time ($\mathbf{s}'$ cannot cause something in $\mathbf{s}$ or any prior state), fully observable (there are no unobserved confounders) and Markov ($\mathbf{s}'$ depends only on $\mathbf{s}$) dynamics, our MG-causal test (Equation 9) this matches a causal discovery test under two conditions: 1) There is not a "true cause" $X_c$ of $X_b'$ with *lesser* information than $S_a$ about $X_b'$, 2) The selection of $do(S_b = \mathbf{s}_b)$ is decorrelated from $S_a$. Here, we use $X_c$ to denote the set of causal variables denoting a cause, and $X_b'$ the set of causal variables denoting an outcome.

In the first case, this describes the case where some factor $c$ confounds $a$ on $b$ that is, $P(\mathbf{s}_b'|\mathbf{s}_b, \mathbf{s}_a, \mathbf{s}_c) \neq P(\mathbf{s}_b'|\mathbf{s}_b, do(\mathbf{s}_a), \mathbf{s}_c)$. Note that because of the Markov assumption $\mathbf{s}_c$ is NOT a common cause of $\mathbf{s}_a, \mathbf{s}_b'$. Instead, this occurs when factor $a$ shares some information with factor $c$, that is $P(S_c|S_a) \neq P(S_c)$, and $\mathbf{s}_c$ is the true cause of the transition dynamics of $\mathbf{s} \rightarrow \mathbf{s}_b'$. In our case, because we choose the factor pair with the highest MG-causal test score, meaning that $S_a$ must be a better predictor of $S_b'$ than $S_c$, *despite* not being the true cause. As an example of this, imagine that in Breakout factor $a$ describes the paddle which rarely interacts with the ball, while factor $c$ is another paddle with noisy observation whose location is determined by factor $a$. In this case, where factor $b$ is the ball and we search for a $c \rightarrow b$ relation, then factor $a$ may confound this information because information about paddle $c$ bounces is subsumed in paddle $a$. This condition is rare but possible in many real-world or real-world-inspired domains, and it is a limitation of using pairwise scope relationships.

In the second case, if $do(a)$ is not selected randomly but based on $\mathbf{s}_b$, and this difference is not accounted for, then the Granger-causal relationship is not captured. This is the case if the input data for $\mathbf{s}_a$ has randomly assigned $\mathbf{s}_a$, for example, if the goals for the policy are sampled randomly and the policy has no dependency on $\mathbf{s}_b$ (which we use in this work). An alternative way to correct this would be to correct values with the importance sampling weights $\frac{P(\mathbf{s}_b|\pi_{\text{directed}})}{P(\mathbf{s}_b|\pi_{\text{random}})}$, where $\pi_{\text{directed}}$ is the policy with imbalanced behavior.

We describe interactions using the language of actual causality because we are implicitly assuming that an interaction occurs when one object is the cause of one behavior in another object in a particular state. In actual causality, the causal relationship is not described in the general case (*could* $a$ cause $b$), but in the specific case between factors (did $a$ cause $b$ in particular state $\mathbf{s}$). The active and passive models also capture

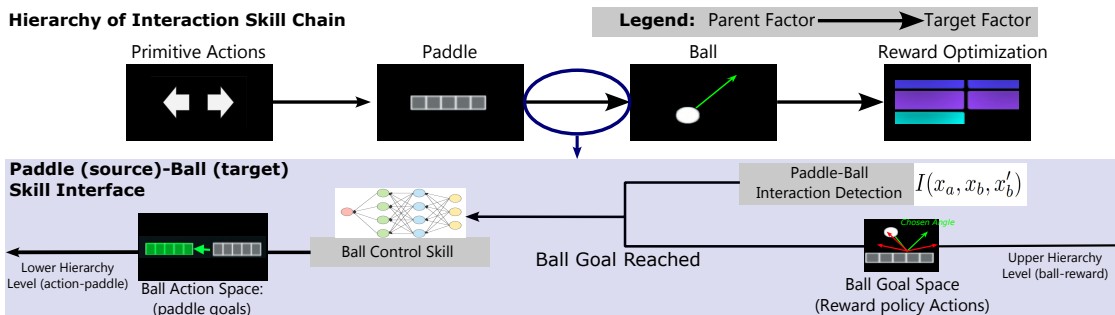

Figure 5: An illustration of the COInS hierarchy. At each edge, a policy controls the target factor to reach an interaction goal, and takes interaction goals as temporally extended actions. The child of the primitive actions passes primitive actions as pseudogoals. **(lower)** Learning a single layer of the HinTS hierarchy, the pairwise interaction between the paddle and the ball. The interaction detector and ball skill policy both use the paddle (parent) and ball (target) state as inputs and indicate if a goal is reached and a new temporally extended action is needed respectively. The ball skill policy uses paddle goals as actions and receives ball bounces as goals from the reward-optimizing policy.

this, because the passive model is trying to capture "what would have happened", and the active model is capturing "but for $a$ in a particular state $\mathbf{s}_a$, resulting in $\mathbf{s}'_b$."

# E    Transferring Skills

When we say that we transfer the learned hierarchy from COInS, this means that we take the final skill, whether ball control by hitting the ball with high accuracy at one of four desired angles in Breakout, or block control by moving the block to a specified location in Robot pushing, and use this goal space as the action space for a high-level policy that takes in all of the environment states. Thus, the action space for this new domain is temporally extended according to the length of time it takes to reach the desired goal or a cutoff. These actions will still call lower level actions (paddle or gripper control), as necessary. In Breakout, the temporally extended skills on the order of ball bounces, which is around 70x less dense, simplify credit assignment, causing performance improvement. We replaced the hierarchy with a known perfect ball-bouncing policy with slightly better results to COInS, supporting this premise. In robot pushing, we suspect that the benefit comes somewhat less from temporal extension, and more from the fact that the action space is much more likely to move the block (compared with primitive actions that move the gripper), which allows for better exploration.

In Breakout, we could have learned an additional layer for block control. However, this would have required added complications to the algorithm: first, it is not always possible to hit a block, either because there are other blocks in the way, or because the block simply does not exist. This would require us to either hard-code a sampler or train one that learns these properties. Next, if we were to transfer this policy, it would then require a policy that chose a block location to hit, either a discrete selection or a continuous location, which would put the onus on the learner to figure out how to hit it. Finally, learning the block policy is sample inefficient, and is likely not to give much benefit, and we can stop learning at any level of the hierarchy. As it is, a block targeting policy is learnable, since the proximity variant essentially captures this.

# F    Network architectures

We use a 1-d convolutional network with five layers: $128, 128, 128, 256, 1024$ which is then aggregated with max pooling and Rectified linear unit activations followed by a 256 linear layer before either actor, critic, passive distribution outputs, and active distribution outputs. In every case, the inputs are the $\mathbf{s}_a$ and $\mathbf{s}_b$ object(s), where we use only a single parent class $a$. In the final evaluation training, $a$ is the set of all factors that are not multiple, except for primitive actions in policy training. We then append $\mathbf{s}_a$ to $\mathbf{s}_b$ and treat each of these as a point. There are also well-known issues in Breakout that without a relative state between the paddle and the ball, the policy fails, so we augment the state with $\mathbf{s}_a - \mathbf{s}_b$ and $\mathbf{s}_b - c$ when appropriate (the

dimensions match). Combined, we call this architecture the Pair-net architecture (a Pointnet architecture for pairs of factors)

We can use the same architecture for all of the policies. Note that when there are not multiple instances of objects (i.e. cases without the blocks in Breakout or the Negative reward regions in Robot Pushing), the Pairnet architecture reduces to a simple multi-layer perceptron. For the sake of consistency, we used the same network for every choice, though this was overparameterized in some cases.

## G  Breakout Training

This section walks through step by step the learning process for COInS in Breakout. Note that the algorithm uses automatic cutoffs to decide which factors to learn a policy over using the interaction score in Equation 9 and comparing it against $\epsilon_{\text{SI}}$, a minimum cutoff weighted log-likelihood. In practice, we found that 5 worked for all edges in Breakout and Robot Pushing, though this can probably be automatically learned based on an estimate of environment stochasticity. The training curves can be seen in Figure 6, and visualizations of the skill spaces in Figure 7.

### G.0.1  Paddle Skill

COInS collects random samples (10000 sample intervals) until it can detect the action-paddle connection through the interaction score (Equation 9). Interaction tests between other factor pairs—action-ball and action-block, have low scores, with details of the comparative performance in Table 4. The paddle interaction detector detects every state as an "interaction" since the actions control the paddle dynamics at every state. $\eta_{\text{paddle}}$ masks out all components except the x-coordinate of the paddle: $[0, 1, 0, 0]$ since it can only move horizontally, and $\eta_{\text{paddle}} \cdot \mathcal{C}_{\text{paddle}'}$ is the x-range of paddle locations. The paddle skill uses $\mathbf{s}_{\text{paddle}}$ as input and learns to perfectly move the paddle to a randomly chosen target x-coordinate in roughly 10k time steps. Paddle training ends when the success rate over 100 updates no longer decreases.

### G.0.2  Ball Skill

COInS continues to gather samples in $10k$ increments using the paddle policy. For every $10k$ additional samples the interaction detectors between the paddle and other factors are updated, and if the performance exceeds $\epsilon_{\text{edge}}$, a skill for that factor is automatically learned. COInS discovers the paddle-ball interaction with the interaction score reported in Table 4 after 80k additional samples. It takes this many time steps because of the infrequency of ball interactions. Using $I_{\text{ball}}$, COInS discovers $\mathcal{C}_{\text{ball}'} \cdot \eta_{\text{ball}}$ with four velocities $[-1, -1]$, $[-2, -1]$, $[-2, 1]$, $[-1, 1]$, the four directions the ball can be struck in Breakout, with $\eta_{\text{ball}} = [0, 0, 1, 1]$. Since $|\eta_{\text{ball}} \mathcal{C}_{\text{ball}'}| < n_{\text{disc}}$, we sample discretely from this set.

The discovered interaction describes a COInS skill trained with hindsight to hit the ball at the randomly sampled angle. Notice that this problem subsumes the one of just playing Breakout, which just requires bouncing the ball. The resulting skill run with random ball velocities plays Breakout well after $< 100k$ time steps as seen in Figure 3 and Table 1, though at that point it only has $\sim 30\%$ accuracy at hitting the ball at the desired angle. Since the success rate has not converged over 100 updates, training continues for $1m$ time steps to achieve 99.5% accuracy for striking the desired angle. At this point, COInS terminates, though future skills such as block targeting were trained as a variant. Since blocks are separate factors, their individual interaction scores are low due to data infrequency. A class-based extension would allow COInS to handle this.

## H  Robot Pushing with NRR Training

The negative reward regions pushing task is difficult, and all these baselines fail as seen in Table 1. Even intrinsic reward methods, like RIDE, struggle in this domain, possibly because the intrinsic reward signal is washed out before the agent can learn to manipulate the block, and sufficient intrinsic reward can be acquired simply by manipulating the gripper. Only HyPE has a non-trivial reward because the options it learns will force it to move the block. However, the skills navigate it to negative reward regions or out of

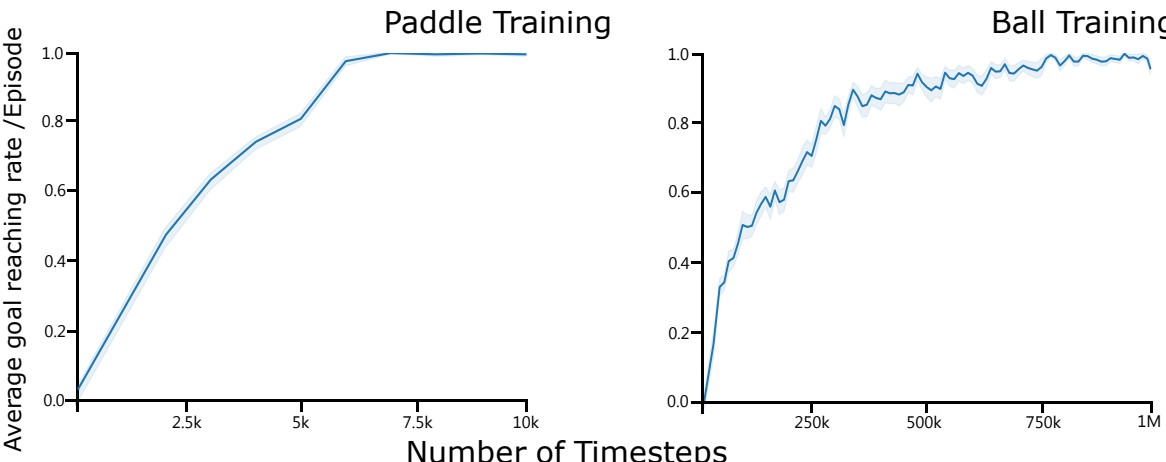

Figure 6: This graph shows the training curves for COInS learning the Breakout skills, where a rate of 1.0 means that the goal is reached on 100% of episodes. We averaged performance over 10 runs. Note the difference in scale between the paddle skill learning (10k time steps), and ball training (10M time steps). Training terminates automatically when performance converges (train goal reaching rate does not increase).
.

bounds most of the time instead of the goal, resulting in a lower trajectory reward than just not moving. In most cases, the agent is highly disincentivized to push the block at all since the likelihood of reaching the goal is low, but the likelihood of entering a negative region is high. However, using a smaller penalty would make directly pushing to the goal optimal—it can take dozens of time steps to push the block around an obstacle. We tried a modified baseline, which pre-trains the agent to push the block to the target, and then fine-tunes the policy to avoid obstacles, but that baseline policy still ends up quickly regressing to inaction. Only COInS achieves non-trivial success, with an average performance of $-21.6$. Importantly, the agent can reach the goal consistently, though it occasionally incurs a penalty for slipping into the negative reward zones because the block skill is agnostic to the NRR. Future work could consider training the policies end to end so that NRR information can be used when learning lower-level skills.

We illustrate the COInS training curves in the Robot Pushing domain in Figure 8 and visualize the goal spaces in Figure 9. We describe the learning procedure here: COInS first gathers 10k random actions according to the same strategy described in Breakout training—it automatically discovers edges using the Interaction test. In this domain, the majority of the benefit arises from based on the amount needed to discover useful correlations with action. It enumerates edges between the primitive actions and the different objects in the domain and learns object models according to the method described in Section 4.

The action-gripper model passes the interaction test with the active model having a weighted performance of 4cm better than the passive model, while the spurious action-block edge fails with less than 0.1cm difference, as enumerated in Table 4. Even though designed for sparse interactions, the model can pick up that the action affects the gripper at almost every time step. The controllable set operation from Section 4.3 returns a mask of $[1, 1, 1]$ since all the gripper components are sensitive to actions. Since the parameter set $c_{\text{gripper}}$ is $> 10$, the goal state is sampled continuously not discretely.

The gripper policy is then trained to reach randomly sampled gripper positions. The gripper model takes as state the state of the gripper and the last action. The gripper policy converges to 100% accuracy at moving within 0.5cm of target positions within 50k time steps. However, again following the $10k$ sampling procedure for gripper-block interaction, we continue to gather gripper movement behavior until gathering a dataset of 100k time steps.

COInS then samples the edge between the gripper and the block, appended with action information as a "velocity" for the gripper. With the combined input state, the forward model passes the interaction test

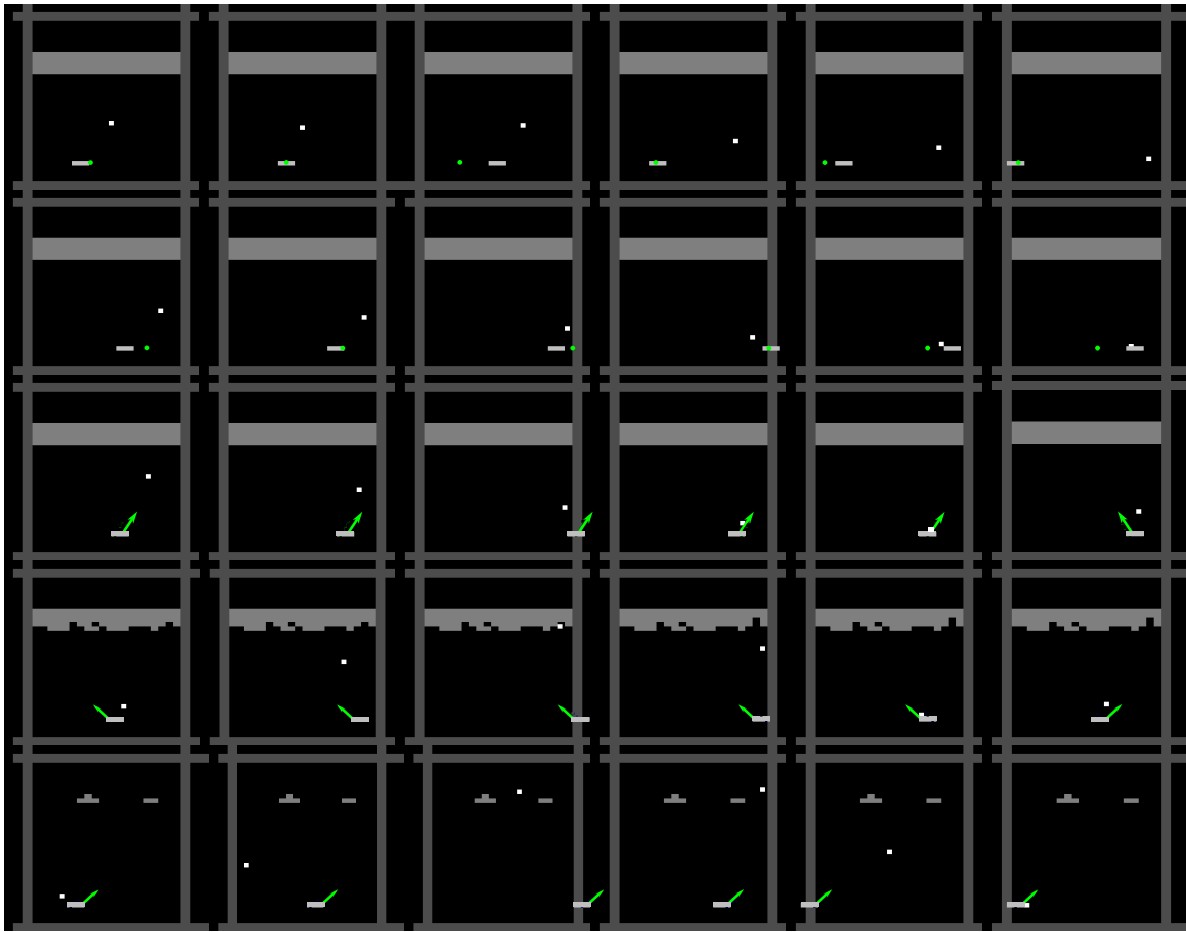

Figure 7: Visualization of the Breakout skills. The first two rows illustrate the block, where the green dot is the desired goal position. The last three rows show the target ball angles, with each row illustrating how the agent manipulates the paddle to produce the desired post-interaction angle. There are four possible angles.
.

with about 1.5cm better-weighted performance. This difference is smaller because block interactions from random actions are rare, and can be low magnitude—the robot only nudges the block. The mask trained over the controllable area is $[1, 1, 0]$ since the block height does not change (the gripper cannot grasp). The block policy learns to move the block to within 1cm of accuracy, and the block-pushing skill converges within 500k time steps.

Learning the final policy with the COInS skills takes around $700k$ time steps. This is because the goal-based movement of the block allows the agent to navigate carefully around objects. However, the task is sufficiently difficult that even with that, the agent still plateaus in performance even though a human using the learned options could get a better reward. This is probably because there are many local minima. This is where a model that predicts block movement could be incredibly useful for learning because it would allow imagining multiple trajectories without getting collapsed into a single low-reward one.

# I    Hyperparameter Sensitivity

In Sections A.1 and A.2 we briefly describe some of the hyperparameter details related to the COInS training process, but in this section, we provide additional context for the sensitivity of hyperparameters described in Table 6 and throughout the work. In general, the volume of hyperparameters comes from the reality that a hierarchical causal algorithm is a complex system. RL, causal learning, and hierarchy all contribute

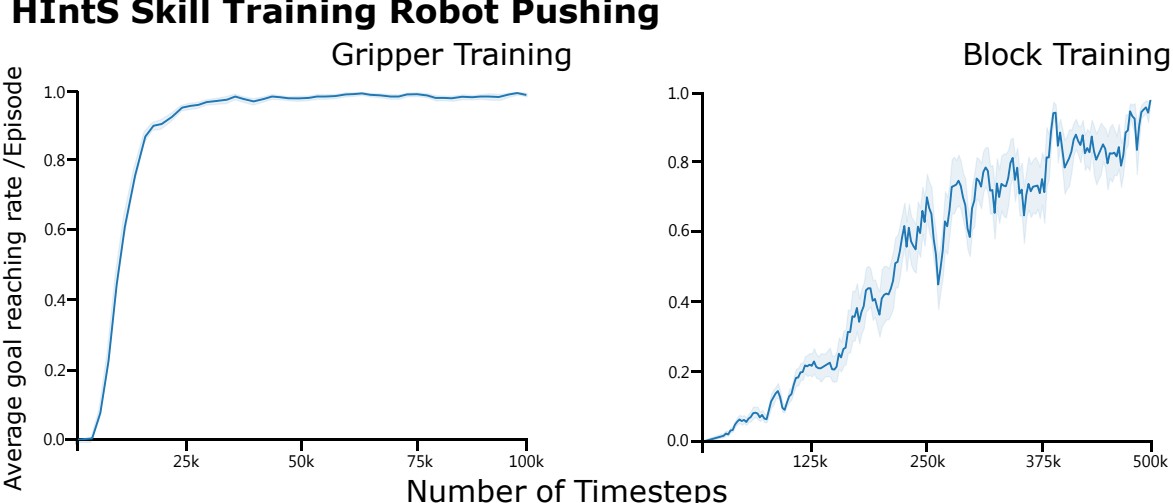

Figure 8: Training curves of COInS learning the Robot pushing skills. **Left**: Gripper reaching. **Right**: Block pushing. 1.0 means a goal is reached (interaction and within epsilon of the desired state) on 100% of episodes, averaged performance over 10 runs. Note the difference in scale between the gripper skill learning (100k time steps), and block training (500k time steps). Training terminates automatically when performance converges (train goal reaching rate does not increase).

.

hyperparameters, and though the overall algorithm may not be sensitive to most of them, some value must be assigned to each. In this section, we aim to provide some clarity on hyperparameter choices.

To construct the skeleton and skills of the skill chain, this algorithm introduces the minimum set size $n_{\text{disc}}$, minimum test score $\epsilon_{\text{SI}}$, success convergence cutoff 0.01, and success convergence timesteps $N_{\text{complete}}$. These terms can be chosen without any tuning since they generally capture details about the environment that often have significant effects. For example, the minimum test score simply needs to be chosen based on how difficult it is to provide accurate predictions of the state. More randomness would suggest a lower value, but this value is just to determine when the algorithm should give up. As a result, the $\epsilon_{\text{SI}} \pm 2$ without causing COInS to exit early. A simple way to generally select this is to train the passive model on the data, and then select the minimum test score to be one minus the average performance of the passive model. Similar to $\epsilon_{\text{pas}}$, in rare-interaction environments this will generally indicate one order of magnitude less likelihood on average prediction for that state factor. Note that a spurious edge such as action→ball in Breakout could never occur regardless of the choice of $\epsilon_{\text{SI}}$, since COInS always chooses the highest likelihood edge, which would be action→paddle. Typically, after the learnable edges have been exhaustive, the difference in likelihood is as much as an order of magnitude for any new edges. $n_{\text{disc}}$ follows a similar reasoning—video games can have discrete outcomes, and so we want to capture that possibility, but the value for cutoff can vary significantly. Since there are usually a limited number of these kinds of choices for a single factor, a choice of 10 is reasonable across many domains.

For the interaction learning hyperparameters: $\epsilon_{\text{pas}}, \epsilon_{\text{act}}, \Sigma_{\text{min}}$, these values are generally environment specific. In particular, since the passive and active epsilons utilize environment information, they should be selected based on what "good" or "bad" prediction is. We convert some of the mean errors into the relevant units for each environment in Table 4. In relevant units, the difference between the predictions of the passive and active models is often significant (6 pixels or 2cm for Breakout and Robot pushing respectively). However, this property does not necessarily hold for all domains. To set $\epsilon_{\text{pas}}$, the simplest solution is simply to train the passive model over the dataset first, then set the $\epsilon_{\text{pas}}$ to be one less than the average value, since this is an order of magnitude lower likelihood event. Alternatively, we have found that 0 appears to work well out of the box. To set $\epsilon_{\text{act}}$, we can similarly use the average performance of the passive model on all the data. When interactions are rare, this will be indicative of good prediction. Overall, the choice of $\epsilon_{\text{pas}}, \epsilon_{\text{act}}$ has a limited effect on the overall structure (which skills to connect together), but must be chosen so that the

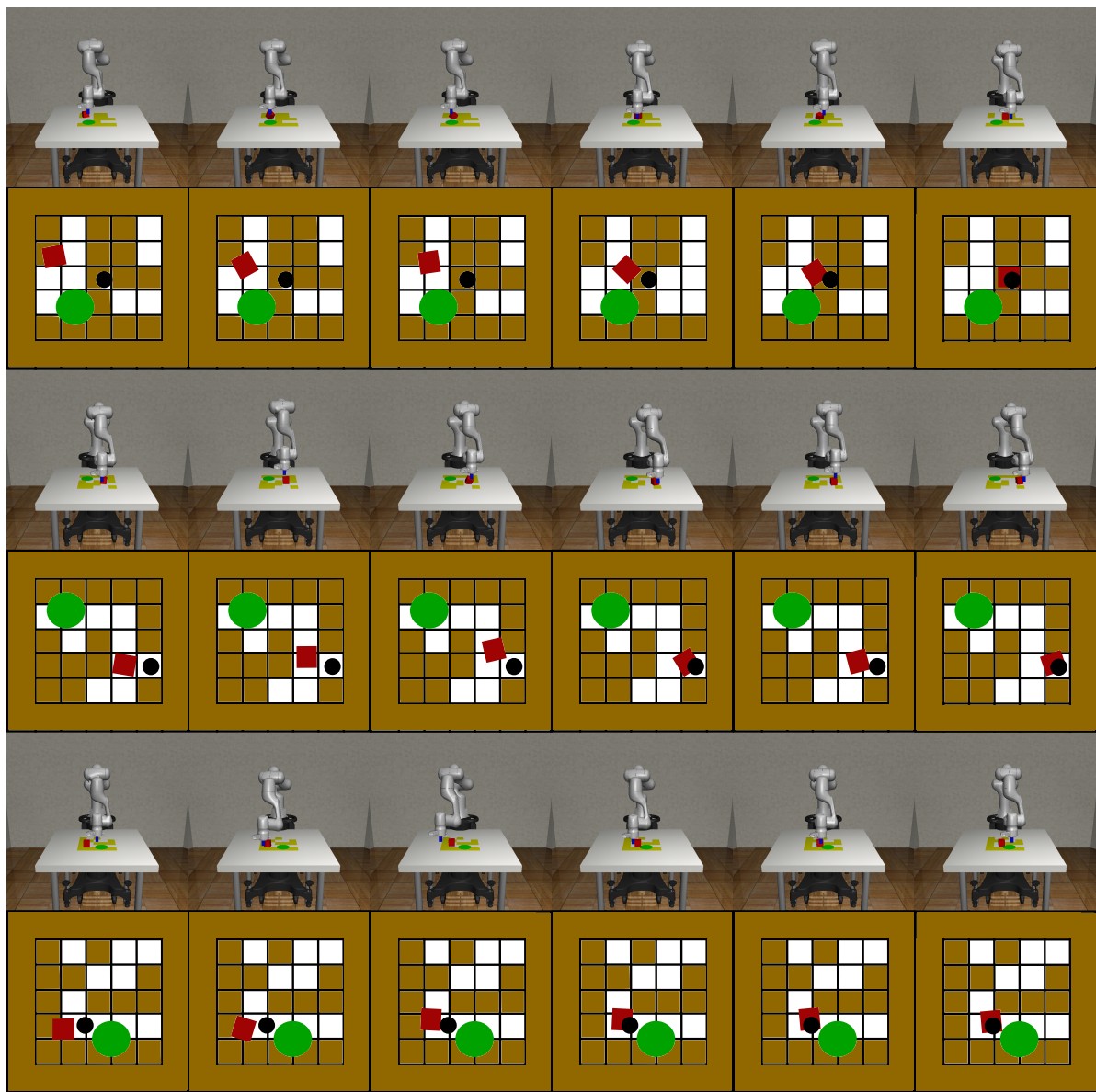

Figure 9: Visualization of the Robot Pushing Block skill, where the upper row renders the robot environment, and the lower row illustrates the state on a grid. The block goal is represented as an x,y position, since the block z coordinate is not controllable, represented on the grid as a black dot (the green circle is the task goal, which relates only to extrinsic reward.

.

number of inaccurately classified interactions is low. If the number of inaccurate interactions is sufficiently high, skill learning can fail to converge. In Breakout and Robot pushing, a choice of $\epsilon_{\text{act}} \pm 1$ and $\epsilon_{\text{pas}} \pm 2$ would not affect the outcome of the connectivity (the edge between the paddle and ball in Breakout, for example). Since these are log-likelihoods, this demonstrates there is a sizable gap between the predictions from the two models on true interaction states. However, in certain states changing the $\epsilon$ might result in more misclassified interactions, which would degrade RL performance. $\Sigma_{\text{min}}$ is the minimum variance of the Gaussian distributions output by $m^{\text{act}}, m^{\text{pas}}$, and setting this to 0.01 seems to work in most domains, but the choice affects what the average log-likelihood will be.

With skill learning parameters: $\epsilon_{\text{close}}, \epsilon_{\text{rew}}, \epsilon_\eta$, hindsight selection rate, and relative action ratio, these are chosen using intuition from the environment except for $\epsilon_{\text{rew}}$, which does require tuning. The environment-specific parameters are often not particularly sensitive since the margins for the environment are often quite substantial. For example, $\epsilon_{\text{close}}$ just needs to be set to a value that is low enough to be a reasonable definition of "reaching a goal." $\epsilon_{\text{rew}}$ on the other hand could have wildly different results because a setting too low would result in no incentive for the agent to search out reward, and a setting too high would result in exploding losses. It might have also been possible to perturb the discount rate $\gamma$ in the RL losses for the same effect, but we used $\gamma = 0.99$ in all skills and reward optimization. $\epsilon_\eta$ is robust in a similar way to values like $\epsilon_{\text{act}}$ and $\epsilon_{\text{pas}}$, with an effective range around $\pm 0.1$ of normalized units (it would have been around $\pm 1$ in log likelihood, but we used the mean difference). A simple way of identifying the appropriate value for this is to use a fixed percentage of the possible range for that feature. If the difference in the range of that feature is greater than 10%, then it is likely to be the result of an interaction, rather than some spurious co-occurrence. Alternatively, a more robust strategy might analyze the errors in the passive model, and select some quantile range for the errors. As an example of how we set the mask in Breakout the ball velocity mask used changes of 0.5 of the possible range ($-2$ to $+2$). from the position changes 0.05% of the possible range (at most 1 or 2 pixels difference of a range of size 84).

In general, regarding hyperparameters, we believe that many of the values can be set automatically, especially based on the average values of the active and passive values, and plan to investigate this in future work. Since the focus of this work is to analyze the performance of Granger causal models when used to construct a skill chain, we primarily focused on finding a usable set of hyperparameters, rather than devising adaptive strategies that can apply across many domains. The intuition for the hyperparameters is often based on inherent stochasticity in the domain, and finding ways to identify the properties of domains is the subject of ongoing research, both in future work related to this project and in the machine learning community.

The RL and network parameters often result in the most difficulty tuning, since they often do not have clear intuitions. However, in general choices such as the continuous and discrete learning algorithms, the learning rate and network architecture did not have a significant effect on the outcome *based on the search of prior work*. That is, without borrowing parameter choices from previous work related to learning rate and algorithm choices, COInS and other baselines, would struggle. However, when building on top of an existing set of RL hyperparameters, performance was stable even when some of these parameters were then perturbed.

As a whole, the majority of hyperparameters have wide margins for their settings, because the difference between Granger models for interactions, or goal-reaching for skills is relatively broad. As a result, the majority of the same parameters apply to both Breakout and Robot pushing, and may also apply to many other domains. It seems likely these could also be automatically identified, though we did not explore this possibility in this work.

## J   Additional Baseline Details

For all the baselines, to mitigate the effect of COInS possibly having an advantage in the state representation, we augmented the state space with relative state features such as the paddle-ball information in Breakout, and the gripper-block and block-goal information in robot pushing. Combined with the Pointnet architecture, this makes the vanilla algorithms object-centric, though they do not have the hierarchical and interaction advantages that COInS is demonstrating.

### J.1   Hypothesis Proposal and Evaluation (HyPE)

The HyPE algorithm relies on discrete action spaces which requires changing the Robot pushing environment by constructing a discrete action space for HyPE to use. This space consists of actions in the cardinal directions x,y, and z, where the agent moves 1/10 of the workspace for each action. This was tested by having a human perform demonstrations with this workspace. HyPE also performs much better in Breakout, where the quasi-static assumptions capture the relationship between the paddle and ball especially well since changepoints capture the instantaneous motion easily. However, in the Robot pushing domain, it struggles

because the block movement can vary in magnitude, but the nature of the actions is such that any movement will get assigned to an action completion. As a result, learning the block-pushing options is only somewhat effective, able to move the block in some regions of space, but getting stuck in others. HyPE is the only baseline that performs meaningful learning on the base task, starting from a $-400$ reward to only $-90$. Ironically, it also has the lowest performance because the other baselines just never touch the block.

HyPE has two varieties, one where the agent is trained using Covariance matrix adaptation evolution strategy (CMA-ES) Hansen et al. (2003) (C-HyPE), and another where it is trained with Rainbow Hessel et al. (2018) (R-HyPE). In the CMA-ES case, while the evolutionary algorithm is good at picking up easy relationships like the ball to the paddle, it cannot learn multiple policies to hit the ball at each desired angle, and only trains to create a bounce changepoint (the next layer that would be used for transfer has only one action, which is why C-HyPE is not trained at transfer). Thus, the CMA-ES version performs well at the main task and poorly at the overall task. On the other hand, because HyPE learns a separate policy for each ball angle, it takes much longer to learn. These policies struggle to keep the ball from dropping (letting the ball go past the paddle), which is the main reason for the poor transfer performance when compared with even the pretrained-fine tuned baseline.

C-HyPE is trained with a 128-activation hidden layer multi-layer perceptron, because it needs an $n^2$ computation in the number of parameters, so using the larger PointNet architecture is infeasible. This is another reason why it struggles to learn complex policies.

By comparison, HyPE and COInS learn similar object-option chains, but because the interaction detector is more robust and the goal-based option is more general, COInS can perform when HyPE cannot. HyPE also makes stronger assumptions about the environment, including the quasi-static assumption and discrete actions. The reliance on discrete modes for high-level skills also limits HyPE.

## J.2 Causal Dyanamics Learning

We ran CDL in every domain (Breakout and Robot Pushing) and all the Breakout variants. It failed in every domain. We hypothesize this is because when the data that it gathers is not particularly meaningful, which is true of random actions in both Breakout and Robot Pushing, it struggles to construct a model that is useful to the agent. Without temporal abstraction, the policies struggle, and this lack of temporal extension is exacerbated by a model that makes inaccurate predictions about key predictions (object interactions). Even with a good model of the system, both these domains have very long time horizons, making them very challenging for on-policy algorithms like PPO (which CDL uses for the model-based component). Random shooting only helps for certain special states (before an interaction), but detecting those states can be challenging, and the rest of the time, it gives back sparse signals. On top of the fact that the rewards are sparse, this is why the domains end up being challenging for CDL. We include a table of performance for CDL in Table 5

We tested the model-learning component of CDL and found that it can detect relationships between difficult-to-model objects like the paddle and the ball or the gripper and the block. However, this takes a large number of time steps ($>500000$), and just because a relationship is detected does not mean that the model is generating useful samples for the policy. In particular, this suggests that having a general model to capture general causality is only so useful in RL tasks. Since COInS creates a model to capture certain rare, but incredibly task-important states, and to identify those well, it ends up being more useful for performance, even though the overall model is probably worse at predicting the next state.

Finally, pairwise interaction tests are attractive in comparison to CDL because in domains with many factors (there are 100 blocks in Breakout) and thus a large state space, this can become prohibitively expensive for CDL. Since CDL uses $P(\mathbf{s}'|\mathbf{s}/x_i)$ to detect causal relationships, where $\mathbf{s}/x_i$ denotes that the factor $x_i$ is cut from the overall state, this results in several models that grow in the number of objects, where each network must include all the objects. We change the state space of Breakout so that each block state only has one value (whether that block exists or not), because otherwise, the cost of this computation would be prohibitively expensive.

### J.3 HAC variants

HAC typically uses the full state of the environment as actions for the higher-level policies and goals for the lower-level policies. However, this means that the HAC higher-level policies need to select complete states to reach, and the HAC low-level policies have to reach those states. This is next to impossible, and that means the hierarchy ends up learning nothing: the low-level states learn nothing because they cannot reach any of the goals, and the high-level states learn nothing because their actions are meaningless.

We mitigate this issue by assigning HAC layers to the objects, essentially emulating what COInS does by having each layer control an entity. In Breakout, this means that HAC has one layer that has block goals/rewards and outputs ball goals, one layer with ball goals and then outputs paddle goals, and one layer to convert paddle goals to action goals. However, this full hierarchy also fails because HAC relies on hindsight that is based on a fixed duration of time. Unfortunately, this often means that the ball interactions appear very rarely in the replay buffer, and the top level using actual rewards has no hindsight targets. The best we could find was with a 2-layer hierarchy that had extrinsic reward as the top-level signal, and output ball velocities to a low-level policy that outputs primitive actions. In this case, outputting upward velocities were somewhat learnable and would give some reward. A similar object-level hierarchy was used for robot pushing, though in this case, there was no clear reason why the layers should fail except that the reward structure is difficult enough that HAC probably needed to pre-learn the block moving policy before doing anything else. However, that would make it the same as COInS in terms of the final structure.

### J.4 RIDE details

Rewarding impact-driven exploration (RIDE) Raileanu & Rocktäschel (2020) uses a learned forward and inverse dynamics model, combined with count-based regularization to provide an intrinsic reward for exploration. We ablated over the RIDE tradeoff parameter (the amount of weight for the intrinsic versus extrinsic reward), the RIDE learning rate (rate to train the forward/inverse models) the network architecture (changing the number of hidden layers, and Pointnet or MLP implementation), and the count-based scaling (rate to reduce the reward for similarly hashed components). We implemented hashing by taking the learned embedding from the forward and inverse models, applying a sigmoid, and then taking the values greater than 0.5 as 1 and less as 0.

RIDE has been applied to visual navigation domains with some success, and in this work, we apply what appears to be the first use of this to a factorized dynamical environment. However, RIDE struggles to perform in both environments. We speculate this is because it relies on the inverse dynamics to provide a significant signal to learn a meaningful embedding. Unlike in pixel-based environments, where the embedding often retains much of the information about the state through the convolutional layers, in the factorized format much of the information can be lost, even with a pointnet, because of the use of fully connected layers. As a result, the RIDE-learned representation quickly converges to the minimal information needed to recover the action. In this case, the forward model is easy to predict, and since the loss is based on the l2 error in the prediction of the next state, without much of the information the agent quickly loses much of the intrinsic motivation. This is true in both Breakout and Robot Pushing, where there are a significant number of objects whose state does not change much (the blocks and obstacles), and whose state is not closely correlated with actions.

As a result, the RIDE loss struggles to provide much exploration bonus. The performance of RIDE mirrors the performance of vanilla RL for the simple reason that the best choice of RIDE reward scaling (after searching through several orders of magnitude of scalings), is the one that has the least negative impact. We think that the slight negative impact might be from the intrinsic reward: in Breakout, once the good behavior has been found (bouncing the ball), the task is closer to exploitation than exploration and overt exploration often reduces performance. For computational reasons, and because there is no reason that an intrinsic reward method such as RIDE should show significantly different transfer than pretraining-fine tuning, we did not run RIDE on the Breakout variants.

| Method | No Proximity | Proximity |
|---|---|---|
| FP | $2.4 \cdot 10^{-3}$ | $2.2 \cdot 10^{-5}$ |
| FN | 0.07 | 0.003 |

Table 3: Table of breakout interaction predictive ability. A false positive (FP) is when the interaction model incorrectly predicts a ball bounce, measured per state, and a false negative (FN) is when the interaction model fails to identify a ball bounce, measured per bounce. We get these bounces from the simulator, so these comparisons are ground truth. These ball bounces are not given to the interaction model during training—we just use them for this evaluation. While both FP, FN are undesirable, FN is a greater issue for learning because they result in missed ball bounces.

| Parent | Act | Act | Pad | Act | Act | Grp |
|---|---|---|---|---|---|---|
| Target | Pad | Ball | Ball | Grip | Blk | Blk |
| PE | 1.03 | 1.21 | 6.81 | 4.8 | 0.22 | 2.2 |
| AE | 3e−3 | 1.69 | 0.30 | 0.17 | 0.18 | 0.6 |

Table 4: Table of interaction test scores (Equation 9) for interaction training in breakout and Robosuite pushing between pairs of objects. "Act" is actions, "Pad" is the paddle, "Grip" is the gripper. "PE" is passive error in pixels for breakout (left) and cm in Robosuite pushing (right), "AE" is the active error in pixels/cm (weighted by interaction). In this table, we convert likelihood comparisons to l2 distance in *cm* using the mean of the normal distribution, which is proportional but not equivalent. We do this because distances are more interpretable.

| Algo | Break | Push | single | hard | big | neg | center | prox |
|---|---|---|---|---|---|---|---|---|
| Base | $85.6 \pm 8.3$ | $30 \pm 0$ | $-5.8 \pm 0.5$ | $-7.1 \pm 0.8$ | $.74 \pm 0.1$ | $2.9 \pm 0.3$ | $-37 \pm 11$ | $0.1 \pm 0.2$ |
| FT | NA | NA | $-5.4 \pm 0.9$ | $-6.1 \pm 0.4$ | $.79 \pm 0.1$ | $-7.0 \pm 19$ | $-42 \pm 9$ | $0.2 \pm 0.1$ |
| HAC | $74.3 \pm 8.2$ | $-43 \pm 89$ | NA | NA | NA | NA | NA | NA |
| RIDE | $92.5 \pm 8.2$ | $-31 \pm 2.2$ | NA | NA | NA | NA | NA | NA |
| CDL | $-49 \pm 1.6$ | $-33 \pm 1.0$ | $-10.0 \pm 0.0$ | $-9.7 \pm 1.5$ | $-9.26 \pm 2.7$ | $-49.9 \pm 2.5$ | $-49.8 \pm 0.6$ | $-10.0 \pm 0.0$ |
| C-HyPE | $76 \pm 15$ | $-21 \pm 5$ | $-6.4 \pm 3.4$ | $-7.3 \pm 1.0$ | $.35 \pm 0.05$ | $-0.72 \pm 0.9$ | $-39 \pm 10$ | $-0.6 \pm 2.5$ |
| R-HyPE | $95 \pm 3$ | $-90 \pm 19$ | $-5.6 \pm 0.45$ | $-5.1 \pm 0.49$ | $.67 \pm 0.06$ | $2.9 \pm 0.46$ | $-20 \pm 1.5$ | $-0.3 \pm 0.3$ |
| COInS | $99 \pm 0.2$ | $-21 \pm 5$ | $-3.2 \pm 1.2$ | $-4.2 \pm 0.9$ | $.85 \pm 0.06$ | $3.6 \pm 0.3$ | $-12 \pm 4$ | $0.5 \pm 0.05$ |

Table 5: Table of the final evaluation of trained policies on Breakout variants COInS trains a high-level controller using the learned options, "FT" fine-tunes a pretrained model, and "Base" trains from scratch. The baselines are evaluated after 5m time steps, while COInS is trained for 2m time steps. The abbreviations are the ones used in Section 5.1

| Parameter | Values | Relative performance |
|---|---|---|
| Continuous Learning Algorithm | SAC, DDPG | similar |
| Discrete Learning algorithm | DQN, Rainbow | similar |
| Learning rate | .0001-.0007 | significant if too high |
| Interaction Proximity | 5-7px, 0.5-1cm | significant if too small |
| $\epsilon_{\text{close}}$, parameter proximity | 1px-2px, 1cm-2cm | significant if too large |
| Network layers | 3-5 | Fixed across skills |
| Network width | 128-1024 | Fixed across skills |
| hindsight selection rate | 0.1-0.5 | moderate sensitivity |
| inline training rate | 100-1000 | sensitive if too low |
| relative action ratio | 0.05-0.3 | moderate sensitivity |
| Constant negative reward $\epsilon_{\text{rew}}$ | $-0.1\text{-}-1.0$ | sensitive |
| Maximum passive log-likelihood $\epsilon_{\text{pas}}$ | 0 | low sensitivity |
| Minimum active log-likelihood $\epsilon_{\text{act}}$ | 2 | moderate sensitivity |
| Feature Mask cutoff $\epsilon_{\eta}$ (normalized units) | 0.1 | low sensitivity |
| Minimum next state distribution variance ($\Sigma$) | 0.01 | moderate sensitivity |
| Minimum Set size $n_{\text{disc}}$ | 10 | low sensitivity |
| Minimum test score $\epsilon_{\text{SI}}$ (in log-likelihood) | 3 | low sensitivity |
| Success Convergence cutoff | 0.01 | Affects sample efficiency vs final performance |
| Skill Convergence timesteps ($N_{\text{complete}}$) | 10000 | Affects sample efficiency vs final performance |

Table 6: Table of hyperparameters

