# OpenReview forum: "Granger Causal Interaction Skill Chains"
_TMLR — Accepted by TMLR_

### Review · Reviewer_JuHF · 2024-01-07

**Summary Of Contributions:**

This paper proposes COInS, that learns a chain of skills using detection between state factors. This provides unsupervised reward signals. The method first detect target and source goal state factors, based on which a reward function is defined at any state. Then the algorithm optimize for an RL policy and construct a chain of skills. The method gives competitive performance on two environments.

**Audience:**

Yes

**Claims And Evidence:**

No

**Requested Changes:**

The idea of using Granger causality for HRL is interesting. However, I find it difficult to understand the techniques developed in this paper as many of the notations are confusing. There also lack detailed explanation of how each component works and their implication. I request the authors to rewrite the methodology part. Some examples:

1.  In section 3.2, should it be $S=S_1\times...,\times S_n$? It took me a while to realize that $\bf{s}_a$ refers to the factor $a$ of the state.

2. In section 3.3 the definition of $\pi$, what does it mean by this policy if the main argument $A_\omega$ is an action set?

3. $C_{\omega_i}$ always in $S$?

4. Equations (3) and (4) are confusing. What does the notation $S\rightarrow \mathcal{N}(\mu, \Sigma)$ even mean? What is $\mu$ and $\sigma$

5. What exactly is the meaning of a passive test? I suggest the authors spend more text on explaining passive test and active set as they are key to your algorithm. How is these function trained?

6. What exactly is "target to control"?

7. In equation (7) why the first term is scaled by $1/D$ while the second is not?

8. How is data collected? And how does the data distribution affect the quality of the algorithm?

9. It is still unclear to me how a chain of skills is learned by the reward function (12). And what exactly is the benefit of your method? Is it learning a better representation for the neural nets that can facilitate downstream tasks?

**Strengths And Weaknesses:**

Strength: The problem is well-motivated. The idea of using casualty mask to devise unsupervised reward signal is interesting.

Weakness: Many of the notations are not clear. The technical contribution lacks clarity which hinders reading. Please see the requested changes for more questions.

---

> ### Author Response · Authors · 2024-01-19
> **Response to Reviewer JuHF**
>
> We appreciate the reader's careful reading of the paper and hope that the adjustment we have made in the updated version will help clarify the notation of the paper, and clarify the technical contribution of the paper. In particular, we have rewritten the methodology portion of the paper and added a component to Figure 2 to clarify the active and passive granger models. We respond and direct the reader towards parts of the updated paper for the specific concerns:
>
> 1) We appreciate the suggestion to change the notation around $\mathcal S$ and changed the notation. We also added additional clarification at the beginning of Section 4 to highlight what $\mathbf s_a$ refers to.
> 2) We adjusted the notation in Section 3 and throughout to refer to $\pi(\cdot|\mathbf s)$, a probability distribution over the set of actions. When this set of actions is $\mathcal A_\omega$, this means that it might be sampling discretely from a set of options, or sampling continuously from a goal space $\mathcal C$. In COInS, we use both of these cases.
> 3) Yes, generally the goal space of the options $C_{\omega_i} \subseteq S$, a subset of the true state space. Since this work focused on factored space goals, in an abuse of notation, $\mathbf c_b$ in section 4.4.1 refers to an assignment of the unmasked features of $\mathbf s_b$, and we updated the work to make note of this in 4.4.1 after equation 11.
> 4) We appreciate this question. The notation $\mathcal S\rightarrow \mathcal N(\mu, \Sigma)$ indicates that the model $m^\text{pas}$ maps states to a prediction of the mean and standard deviation of a normal distribution of the prediction of $\mathcal x_b'$, the next target state. We have edited the surrounding sections to make this clear.
> 5) We have updated the work to improve the clarity describing the passive and active models, by adding Figure 2a (a diagram describing the passive and active modeling inputs and outputs, and clarifying that the methods are trained to maximize log likelihood (the sentence before Equations 4, 5). We have hopefully also removed notation-related confusion around these questions.
> 6) We have clarified that source and target are terms used to describe the head and tail of a pairwise relationship, so that if we are training paddle$\rightarrow$ball, the paddle is the source, and we are learning to control the ball state (by learning to induce goal velocities post-interaction). Where this phrase occurs in the work we have worked to clarify the terms.
> 7) The missing scale is a type, and we have corrected it in the updated version
> 8) We appreciate this clarification. We use the data collected during the RL training of previous iterations of the COInS loop as the data to train the models. Indeed, the data distribution does affect the quality of the algorithm, but this data is self-induced. For the first edge, we use random action data and repeatedly train until we can find an edge (described in Section 4.5)
> 9) The reward function in Equation 12 does not learn a chain of skills, but a single skill. For example, the first interaction in Breakout searches for a relationship between the primitive actions as the source factor and the other factors using the interaction scores chooses one target factor and learns a skill relating the primitive actions to the paddle using the reward function in 12, with $h_{\text{prim, paddle}}, C_\text{paddle}$. Then, using the paddle as the source object, it repeats the search, identifies another target factor (the ball), and then applies the reward function in 12 with $h_{\text{paddle, ball}}, C_\text{ball}$.

---

> ### Comment · Reviewer_JuHF · 2024-01-23
> **acknowledge of authors' response**
>
> Thanks to the authors for addressing my questions and making several revisions. I do not have more questions at the current moment, but I would appreciate it if the authors could further clarify the algorithmic whole pipeline. E.g., how does the unsupervised reward learning interact with the chain of skills & HRL. This is not perfectly reflected by Figure 2 and the algorithmic diagram.

---

> > ### Author Response · Authors · 2024-01-24
> > **Clarification of algorithmic pipeline**
> >
> > We are glad that the revisions and response helped to clarify some questions and we hope that we can make the algorithmic pipeline easy to understand.
> >
> > The algorithm works as an iterative loop, where each iteration learns a skill for controlling one of the factors until the score functions indicate that there are no more factors to control. The unsupervised reward is used to train each skill during the loop, but differs based on which variables $a,b$ are being used. As in the paper, we will use the example of Breakout.
> >
> > Iteration 1 of the loop takes the primitive actions and uses a dataset of random actions (10000 initial states), to determine the interaction relationships. So it tests ($Sc_I(D, \text{primitive actions}, b)$) for a relationship between primitive actions and the paddle, primitive actions and the ball, primitive actions and the blocks, primitive actions and reward, etc. The paddle has the highest score $Sc_I(D, \text{primitive actions}, \text{paddle})$ and a score greater than the threshold, so that is used as the target object. Then a policy is trained with reinforcement learning which takes in a goal state $\mathbf c_\text{paddle}$ a position for the paddle and the interaction model for the paddle $h_{\text{primitive actions}, \text{paddle}}$, and finds the sequence of primitive actions necessary to reach the desired state. Notice since the actions always control the paddle's next position, $h_{\text{primitive actions}, \text{paddle}} = 1$ everywhere. Thus, the goal-based RL loss is $-\epsilon_\text{rew}$ everywhere, except when the paddle is at the location $\mathbf c_\text{paddle}$. When training, $\mathbf c_\text{paddle}$ are sampled randomly from the range of locations that the agent has seen before (in the initial random dataset). It takes about $10000$ additional states to train the paddle to reliably reach an arbitrary sampled $\mathbf c_\text{paddle}$
> >
> > Iteration 2 of the loop takes the paddle state and uses the dataset up to that point (10000 random action states, 10000 paddle training states), to learn a new interaction relationship. This time, it tests $Sc_I(D, paddle, b)$, and finds the ball has the highest score $Sc_I(D, \text{paddle}, \text{ball})$. This time, $h_{\text{paddle}, \text{ball}}$ is mostly $0$ since the ball rarely interacts with the paddle, but goal-based RL works the same way: the agent receives $-\epsilon_\text{rew}$ reward everywhere, except when $h_{\text{paddle}, \text{ball}}=1$ and the ball velocity at that state is equal to $\mathbf c_\text{ball}$, which in practice corresponds to a particular bounce angle. However, unlike with the paddle skill, the action space for the ball policy is temporally extended, the space of $\mathcal C_\text{paddle}$.
> >
> > Iteration 3 of the loop takes the ball state and uses the dataset up to that point (10000 random action states, 10000 paddle training states, 1M ball training states), to find new interaction relationships. This time, it tests $Sc_I(D, ball, b)$, and does not find any $b$ such that $Sc_I(D, ball, b) > \epsilon_\text{SI}$. This is mostly because the blocks are treated as separate entities, so no single block is hit often enough to get statistically significant scores. Since there is no clear edge to add, the COInS loop terminates. However, if we used a class-based representation of blocks, the chain could probably have continued.
> >
> > As we mention in the paper, this iterative loop progresses automatically based on the scores in the same pattern for different domains. In summary, the Granger-causal interaction functions serve two primary purposes: first, to evaluate $Sc_I(D, a, b)$ to determine which $a$s and $b$s to connect as links in the HRL chain. Second, to define the unsupervised reward function, where there is only nonnegative reward when $h_{a,b} = 1$. The same structure of reward is used for every level of the hierarchy, but since $h_{a,b}, \mathcal C_b$ will differ at every iteration, the specific reward will differ.

---

### Review · Reviewer_TzJM · 2024-01-08

**Summary Of Contributions:**

This paper introduces COInS, a hierarchical reinforcement learning method for factored state spaces. COInS uses a test based on the Granger causality test to build a chain of controllable factors, where the goals of a child factor are the actions of the parent factor.

The authors provide experimental results for the proposed method in Atari Breakout and a custom block-pushing problem in the physics-based Robosuite, compared to a number of baseline and basic RL methods. The authors also look at transfer learning in a direct and fine-tuned setup to modified versions of the training domain.

**Audience:**

Yes

**Broader Impact Concerns:**

Not applicable.

**Claims And Evidence:**

Yes

**Requested Changes:**

All the issues / questions below are relatively small concerns:


“Unlike reward-based skills, humans, even as young as infants, exhibit exploratory behavior without top-down rewards.”
Possibly weaken this statement very slightly, maybe something like “without obvious top-down rewards”.  Otherwise, I’m not sure that’s inarguably true: there are arguments made for humans as an agent having a complicated reward function, with some preference for novelty for its own sake in some more abstract features.

eq’n 3 and 4 – should script N and N be the same?

Eq 5 and 6, log likelihood, and negative log likelihood as loss.
As the text stands, it actually seems to be saying the eq’s 5 and 6 are negative log likelihood. It might be more clear to say “we train m^pas and m^act to maximise the log likelihood of the observed data: eq 5 and 6”.  If you really want to be clear you’re minimising negative log-likelihood add something like “(by using \negative log-likelihood as the loss function.)”
Related, the following paragraph about a high Markov-Granger score would be more clear without flipping back and forth between loss.

section 4.3: “With l as define is Equations 5 6”  5 and 6

1 - epsilon_act:  Why the choice of 1-epsilon for a log likelihood, rather than just -epsilon or even epsilon?  This seems like it’s pointing at probabilities instead of log likelihood, where there’s a natural sense where 1-p is meaningful.

“states with high passive error log f”
what is f?

“In cases (1) and (2), l(s_b,s_a,s’_b;psi) will be high”.
As earlier, with log-likelihood and loss=negative log likelihood, it would be more clear if the text avoided flipping between views with different signs: l_pas where low is a low probability prediction, “predicts with low accuracy” where low is a low probability prediction, and “high passive error” where low is a high probability prediction.
This leads to “In cases (1) and (2) l_act(...) will be high”.  The probability l_act / accuracy of m^act will be low when the environment is highly stochastic or a is not a relevant causal factor. The loss – which is not l – will be high.

“When searching for general MG-causal relationships, the score can often be”
What score? A causal score hasn’t been introduced yet, only the non-causal score of eq’7.

“Where N_int represents is the number of detected interactions in D”    (remove “is”)
Define N_int. It’s presumably sum_D h(), but the reader shouldn’t need to presume.

Eq 11
Is this work assuming features are in [0,1]? Section 3.2 only mentions “a fixed length vector of real values we call features”.  However, the use of ||s-u|| > 1-epsilon seems to suggest that the features are in a [0,1] range.

“then the goal space is discrete, otherwise, it is continuous”
“... the goal space is treated as if it is discrete, otherwise it is treated as continuous" ?

Check the use of citations throughout: there are a number of sentences like “The goal-based RL used to train our goal-reaching skills utilizes hindsight experience replay Andrychowicz et al. (2017)” The citet citation should either be in the sentence like “.. experience replay of \citet{Andrychowicz2017}” or a parenthetical like “... experience replay (\citep{Andrychowicz2017})”

HIntS – earlier choice of name?  There are a few occurrences of this name.

What are the choices of epsilon_act and epsilon_pas used in the experiments?

Center obstacles: what is the penalty? Is there a reward (+1?) for breaking a non-obstacle block?

**Strengths And Weaknesses:**

The proposed method COInS is able to automatically discover useful relationships in the two test domains, with performance that is roughly equal or better than other hierarchical methods, either in training time or reward.

Given free choice of environments, it would be nice to see a slightly larger difference in performance: rainbow and SAC were strong contenders in all of the experiments. I also wondered how much success comes from the combinations of hand-generated factors, and parameter choices governing the causality tests. Could the tuning of epsilon_act, epsilon_pas, and epsilon_SI (and other parameters) be inadvertently acting as a selection of the skill chain?

That said, it is good that COInS does at least do better than baseline RL in the chosen domains, given a number of existing HRL methods did not.

---

> ### Author Response · Authors · 2024-01-19
> **Response to Reviewer TzJM**
>
> We appreciate the reviewer's careful reading of the work and hope the revising of the writing addresses the concerns and questions.
>
> We first address the primary concern about hand-generated factors. Achieving skill learning directly from pixels is one of the major goals for future work. This work is focused primarily on introducing the efficacy of skill chains constructed using Granger-causal interactions to perform complex, long-horizon tasks. It focuses on introducing the Granger-causal interaction metric and the skill chain architecture rather than feature learning. To ensure fairness, all baselines are given access to the factors, and use the same factor-centric Pointnet architectures that COInS uses. We also provided some additional relative state information, such as the relative gripper-block state representation in robot pushing and the relative paddle-ball representation in Breakout to improve baseline performance. We added details describing this in Appendix J (on baselines). As a result, while some of the improvements can be attributed to the state space (COInS takes advantage of the skills being pairwise), these advantages are somewhat specific to COInS. We tried hard-coding these into HAC, but HAC struggled in general so it was not clear hard-coding state information provided much benefit.
>
> Addressing the concern about hyperparameters, we believe that there are reasonably systematic ways to identify many of the hyperparameters, especially by characterizing some properties of the environment (the normalizing range of possible values for certain features), or by using the trained passive models to identify the inherent stochasticity. We direct this reviewer to the response to reviewer xrRe and Appendix I for a more detailed discussion. We intend to incorporate domain identification methods in future work to automatically specify the majority of these features, such as $\epsilon_{SI}$ and $\epsilon_\text{pas/act}$, but in this particular work we are primarily interested in introducing Granger-causal interactions and describing how they can be used to construct skill chains.
>
> To directly speak to the question of whether the skill chain would be constructed differently given a different setting of the parameters $\epsilon_{\text{SI}}$ and $\text{Sc}_I $ can only cause the skill chain to stop early, rather than creating different connections. In many domains, the intuitive connections are characterized by the Granger-causal test because it analyzes the time series for statistical significance.  For example, an edge from the actions$\rightarrow$ball could not have achieved a better $ \text{Sc}_I $ score without trivializing the selection of the passive and active thresholds (such as by making them so large/small so that nothing is considered an interaction). We also note that because of the relatively wide margins for the summary statistics, we are looking into selecting these values adaptively rather than choosing fixed values.
>
> We greatly appreciate the reviewer reading through the work and noticing several of these concerns, and we provide brief notes about the small concerns here:
> - ``unlike...'': we've made the suggested adjustment to weaken this statement
> - equations 3 and 4 should use the same $\mathcal N$
> - We greatly appreciate the reviewer pointing this out and have adjusted the paper to describe only log-likelihood
> - 4.3: comma added
> - $\epsilon_\text{act}$ is unbounded, so we have adjusted this accordingly
> - $f$ is replaced with $\ell$
> - Similarly to the other point about log-likelihood, this has been resolved.
> - The score for MG-causal is still causal, just not adjusted for interactions, but we've clarified this distinction in the paper
> - $N_\text{int}$ We've clarified this by explicitly defining this value
> - We do not assume normalized features, and adjust this equation accordingly
> - ``treated as discrete'': we changed this wording.
> - We've adjusted a number of citations to the appropriate parenthetical form
> - We've replaced HIntS with COInS
> - In Table 6 (hyperparameters) in the appendix, we note hyperparameter choices. we used 0,2 for all choices of $\epsilon_\text{act}$ and $\epsilon_\text{pas}$
> - In the center obstacles there is a penalty for bouncing the ball off the paddle. There is only a reward when breaking a block, so hitting the blocks in the center will give -1 reward if the agent must bounce the ball off the paddle to keep it in play. This forces it to break the blocks on the sides.

---

> > ### Comment · Reviewer_TzJM · 2024-01-22
> > **Reviewer acknowledgement of respone.**
> >
> > Thanks to the authors for the answers.
> >
> > There were a couple of reviewer comments about the selection of hyperparameters and the sensitivity to the values. I think the inclusion of an appendix discussing this is a good choice. I have no further questions or comments.

---

### Review · Reviewer_xrRe · 2024-01-12

**Summary Of Contributions:**

This paper studies the problem of skill discovery in factorized MDPs and suggests a novel method based on Granger Causality. The proposed CoInS algorithm  identifies causal links between source and target factors and then learns how to control these target factors. CoInS improves over baselines on two different control tasks both w.r.t. sample complexity and performance. The authors also illustrate that the skills learned by CoInS can transfer to new tasks.

I think the main contributions of the paper are:
 -  Using Granger Causality to find interactions between factors.
-  Introducing the algorithm CoIns which identifies interactions between factors and then learn goal-conditioned policies to control these factors.

**Audience:**

Yes

**Broader Impact Concerns:**

No concerns.

**Claims And Evidence:**

Yes

**Requested Changes:**

# Critical:
-	Improve writing and fix the typos, missing definitions etc.
-	Discussion regarding hyperparameters (See weaknesses).

# Other
-	I suggest that the authors cite the conference or journal version of articles and not the arXiv versions. Some examples are Sharma 2019 and Wang 2022.

**Strengths And Weaknesses:**

# Strengths
-	The idea of using Granger Causality for skill discovery seems novel and the authors show that the proposed method has an advantage over baselines.
-	The conceptual idea behind the method is well-motivated by a simple example (Breakout).
-	I appreciate the discussion around baselines, and it helps position the proposed algorithm in relation to already existing ones.
-	The transferability experiments are convincing.

# Weaknesses:
-	The main weakness of this paper is the presentation. I get the feeling that the paper was written in a hurry. Some variables are never defined, such as $\mathcal{X}_b$. Plots and tables contain several typos, e.g. the algorithm HIntS is never defined? I assume this is supposed to be COInS?
-	It is unclear how robust the algorithm is to perturbations of the many hyperparameters, e.g. $\epsilon_s, \epsilon_{close}, \epsilon_{SI}, \epsilon_\eta, n_{disc}, Sc_\min$. Many of these parameters are thresholds where certain parts of the algorithm depends on quantities being above/below the corresponding threshold.  I’m worried that finding the right skills is very sensitive to the choice of thresholds. Some discussion about the robustness w.r.t. changes in these parameters and how hard it is to find the right ones would be nice to see. Also, is it possible to tune them during training?

---

> ### Author Response · Authors · 2024-01-19
> **Response to Reviewer xrRe (1)**
>
> We appreciate the reviewer for their insightful summary of the paper, and have made several passes to improve the primary concern with the writing and typos. Below, we also directly address some of the main weaknesses, and not where in the updated version these questions have been commented on.
>
> Variables never defined: We changed notation from a previous version of the paper, and apologize for the typos and mistakes that occur from the old notation persisting. We have made several careful passes over the paper so that the plots and tables contain the correct notation (HIntS has been removed, and any occurrence of $\mathcal X$ has been replaced by the defined $\mathcal S$, along with adjustments to the tables). We have made changes throughout the methods section and notation, so we did not highlight every change, but we believe that after these modifications the clarity of the work and writing is improved.
>
> We agree that greater discussion of the hyperparameters is warranted, and have included a reference in the paper in Section 4.3 in on page 6 to an added Appendix I going into some detail on the hyperparameters and their tuning. We direct the reviewer there for a more in-depth discussion, but we wanted to directly address some of the hyperparameters here:
>
> $\epsilon_\text{close}$: This value just needs to be set to a domain dependent value based on what constitutes satisfactory goal-reaching. So in Robot pushing that was around $3$mm and in Breakout, that was $0$px, since we had exact positions. Changing those values would affect RL, since the skills would be less reliable.
>
> $\epsilon_\text{act/pas}$: In relevant units, the difference between the predictions of the passive and active models is often significant (6 pixels or 2cm for Breakout and Robot pushing respectively). However, this property does not necessarily hold for all domains. To set $\epsilon_\text{pas}$, the simplest solution is simply to train the passive model over the dataset first, then set the $\epsilon_\text{pas}$ to be one less than the average value, since this is an order of magnitude lower likelihood event. Alternatively, we have found that 0 appears to work well out of the box. To set $\epsilon_\text{act}$, we can similarly use the average performance of the passive model on all the data. When interactions are rare, this will be indicative of good prediction. Overall, the choice of $\epsilon_\text{pas}, \epsilon_\text{act}$ has a limited effect on the overall structure (which skills to connect together), but must be chosen so that the number of inaccurately classified interactions is low. If the number of inaccurate interactions is sufficiently high, skill learning can fail to converge. In Breakout and Robot pushing, a choice of $\epsilon_\text{act} \pm 1$ and $\epsilon_\text{pas} \pm 2$ would not affect the outcome of the connectivity (the edge between the paddle and ball in Breakout, for example). Since these are log-likelihoods, this demonstrates there is a sizable gap between the predictions from the two models on true interaction states. However, on certain states changing the $\epsilon$ might result in more misclassified interactions, which would degrade RL performance.

---

> > ### Author Response · Authors · 2024-01-19
> > **Response to Reviewer xrRe (2)**
> >
> > $\epsilon_\text{SI}$, $n_\text{disc}$: The minimum test score simply needs to be chosen based on how difficult it is to provide accurate predictions of the state. More randomness would suggest a lower value, but this value is just to determine when the algorithm should give up. As a result, the $\epsilon_\text{SI}\pm 2$ without causing COInS to exit early. A simple way to generally select this is to train the passive model on the data, and then select the minimum test score to be one minus the average performance of the passive model. Similar to $\epsilon_\text{pas}$, in rare-interaction environments this will generally indicate one order of magnitude less likelihood on average prediction for that state factor. Note that a spurious edge such as action$\rightarrow$ball in Breakout could never occur regardless of the choice of $\epsilon_\text{SI}$, since COInS always chooses the highest likelihood edge, which would be action$\rightarrow$paddle.
> >
> > We changed notation from $Sc_\text{min}$ to $\epsilon_{SI}$, so these represent the same parameter and we have resolved this in the paper. $n_\text{disc}$ follows a similar reasoning---video games can have discrete outcomes, and so we want to capture that possibility, but the value for cutoff can vary significantly. Since there are usually a limited number of these kinds of choices for a single factor, a choice of 10 is reasonable across many domains.
> >
> > $\epsilon_\text{eta}$: $\epsilon_\eta$ is robust in a similar way to values like $\epsilon_\text{act}$ and $\epsilon_\text{pas}$, with an effective range around $\pm 0.1$ of normalized units (it would have been around $\pm 1$ in log likelihood, but we used the mean difference). A simple way of identifying the appropriate value for this is to use a fixed percentage of the possible range for that feature. If the difference in the range of that feature is greater than $10\%$, then it is likely to be the result of an interaction, rather than some spurious co-occurrence. Alternatively, a more robust strategy might analyze the errors in the passive model, and select some quantile range for the errors. As an example of how we set the mask in Breakout the ball velocity mask used changes of $0.5$ of the possible range ($-2$ to $+2$).
> >
> > In general, regarding hyperparameters, we believe that many of the values can be set automatically, especially based on the average values of the active and passive values, and plan to investigate this in future work. Since the focus of this work is to analyze the performance of Granger causal models when used to construct a skill chain, we primarily focused on finding a usable set of hyperparameters, rather than devising adaptive strategies that can apply across many domains. The intuition for the hyperparameters is often based on inherent stochasticity in the domain, and finding ways to identify the properties of domains is the subject of ongoing research, both in future work related to this project and in the machine learning community.
> >
> > We appreciate the suggestion and have corrected citations where possible to the journal/conference versions.

---

> > > ### Comment · Reviewer_xrRe · 2024-01-22
> > > **Acknowledgement of respone**
> > >
> > > Thank you for your thorough response. The discussion about the sensitivity of the hyperparameters looks good.  My requested changes have been addressed and I have no further questions or comments at this point.
> > >
> > > Best regards,
> > >
> > > Reviewer xrRe

---

### Decision · Action_Editor_FspD · 2024-02-06

**Recommendation:** Accept as is

**Comment:**

This paper proposes a hierarchical RL algorithm that discovers a chain structure in decision making using Granger causality. The paper was rejected by TMLR in 2023 because it was not written clearly. The notation was highly ambiguous even when I read the paper. The revision was reviewed by the same reviewers. The reviewers still pointed out errors and typos, which were promptly corrected by the authors. This was mostly because the authors changed notation to make the paper clearer. All three reviewers support the acceptance of this paper and so do I.

**Audience:**

Yes. This paper would be of a general interest to RL and robotics communities. Moreover, skill discovery is akin to structure learning and thus of a general interest to the machine learning community.

**Claims And Evidence:**

Yes. The proposed method is evaluated on two RL domains: Breakout and Robot Pushing. The reviewers specifically comment on the fact that the experiments are sufficient.